# Early life stress disrupts intestinal homeostasis via NGF-TrkA signaling

Hoi Leong Xavier Wong[1], Hong-yan Qin[2], Siu Wai Tsang[3], Xiao Zuo[4], Sijia Che[1], Chi Fung Willis Chow[1], Xi Li[5], Hai-tao Xiao[6], Ling Zhao[1], Tao Huang[1], Cheng Yuan Lin[1], Hiu Yee Kwan[1], Tao Yang[7], Frank M. Longo[7], Aiping Lyu[3] & Zhao-xiang Bian [ID] [1]

Early childhood is a critical period for development, and early life stress may increase the risk of gastrointestinal diseases including irritable bowel syndrome (IBS). In rodents, neonatal maternal separation (NMS) induces bowel dysfunctions that resemble IBS. However, the underlying mechanisms remain unclear. Here we show that NMS induces expansion of intestinal stem cells (ISCs) and their differentiation toward secretory lineages including enterochromaffin (EC) and Paneth cells, leading to EC hyperplasia, increased serotonin production, and visceral hyperalgesia. This is reversed by inhibition of nerve growth factor (NGF)-mediated tropomyosin receptor kinase A (TrkA) signalling, and treatment with NGF recapitulates the intestinal phenotype of NMS mice in vivo and in mouse intestinal organoids in vitro. Mechanistically, NGF transactivates Wnt/β-catenin signalling. NGF and serotonin are positively correlated in the sera of diarrhea-predominant IBS patients. Together, our findings provide mechanistic insights into early life stress-induced intestinal changes that may translate into treatments for gastrointestinal diseases.

[1] Institute of Brain and Gut Axis (IBAG), Centre of Clinical Research for Chinese Medicine, School of Chinese Medicine, Hong Kong Baptist University, Kowloon Tong, Hong Kong SAR, China. [2] Department of Pharmacy, First Hospital of Lanzhou University, 730000 Lanzhou, China. [3] School of Chinese Medicine, Hong Kong Baptist University, Kowloon Tong, Hong Kong SAR, China. [4] School of Pharmacy, Lanzhou University, Lanzhou 730000, China. [5] Department of Gastroenterology, Peking University Shenzhen Hospital, 518035 Shenzhen, China. [6] School of Pharmaceutical Sciences, Health Science Center, Shenzhen University, 518060 Shenzhen, China. [7] Department of Neurology and Neurological Sciences, Stanford University School of Medicine, Stanford, CA 94305, USA. These authors contributed equally: Hoi Leong Xavier Wong, Hong-yan Qin. Correspondence and requests for materials should be addressed to Z.-x.B. (email: bzxiang@hkbu.edu.hk)

Chronic exposure to adverse life events, like poverty and lack of parental care, imposes detrimental impacts on health and increases risks for functional gastrointestinal disorders, such as irritable bowel syndrome (IBS), later in life[1–4]. Neonatal maternal separation (NMS) in rodents, a well-documented animal model for early-life stress, indeed induces various gastrointestinal dysfunctions, including hyperalgesia to colorectal distension, increased colonic mucosal permeability, and enhanced colonic motility[5–7]. Therefore, NMS is accepted as an experimental model for IBS though it does not completely recapacitate human IBS phenotypes[6–8]. Despite the significant association between early-life stress and gastrointestinal disorders, the mechanism by which early-life stress alters the intestinal homeostasis remains poorly understood.

The hypothalamic–pituitary–adrenal (HPA) axis is important for regulating the homeostatic response to stress. Emerging evidence reveals that the interplay between the HPA axis and nerve growth factor (NGF) plays a crucial role in the development of early-life stress-associated gastrointestinal disorders[9,10]. Acute or chronic stress promotes long-term alterations of corticotrophin-releasing factor (CRF), a key mediator in the HPA axis, in both the central nervous system and intestinal tissues, which in turn increases the expression of NGF in the intestinal mucosa and enhances the release of NGF from intestinal mast cells[11]. Conversely, NGF has been recently suggested to exert stimulatory actions on the HPA axis[12–14]. NGF is a neurotrophic factor that is essential for neuronal development in the nervous system. It is also involved in the regulation of various biological processes in non-neuronal cells, such as pancreatic beta cells and immune cells[15,16]. NGF mediates its major biological functions through tropomyosin kinase receptor A (TrkA). NGF-mediated TrkA signaling has been implicated in the development of inflammation-associated visceral hyperalgesia[17]. Moreover, we and other studies previously demonstrated that the expression of NGF and its cognate receptor TrkA is significantly elevated in both spinal cords and colons of NMS-treated rats[18,19]. Systemic treatment with recombinant NGF during the neonatal stage leads to a wide range of intestinal phenotypes, such as visceral hypersensitivity and disruption of the mucosal barrier, that can be observed in NMS-treated rats and human IBS patients[19,20]. In contrast, inhibiting NGF signaling by the administration of either NGF antagonists or anti-NGF antibodies effectively alleviates the NMS-induced bowel disorders[19,20]. These reported findings suggest that NGF-mediated signaling contributes to NMS-induced bowel dysfunctions. More importantly, there is an upregulation of NGF and TRKA in colonic mucosal tissues from IBS patients[21,22], suggesting the relevance of NGF/TrkA signaling in functional intestinal disorders. Although the central role for NGF signaling in early-life stress-induced intestinal dysfunctions has been suggested, the precise function of NGF signaling in the regulation of intestinal homeostasis in response to early-life stress remains unexplored. Further studies to dissect the function of NGF in the maintenance of intestinal integrity are required to determine the therapeutic potential of targeting NGF signaling in early-life stress-associated bowel disorders.

To maintain intestinal homeostasis, the intestinal epithelium that functions as a physical barrier against enteric pathogens and facilitates dietary absorption is continuously renewed and repaired throughout life, which is driven by intestinal stem cells (ISCs) residing in intestinal crypts. During cell division, ISC not only maintains itself by self-renewal, but it also gives rise to all differentiated intestinal cell types, including enterocytes, goblet cells, enteroendocrine cells, and Paneth cells[23]. ISC is therefore important for the maintenance of intestinal homeostasis. Enterochromaffin (EC) cells are a major population of enteroendocrine cells in the epithelial lining and form the primary site for the synthesis and release of serotonin. In the gastrointestinal tract, serotonin released from mucosal EC cells activates neural reflexes to regulate intestinal motility and secretion[24]. EC cell hyperplasia and deregulated production of serotonin from EC cells have been found in colons from IBS patients[25,26]. Moreover, the treatment with 5-HT3, a specific serotonin antagonist, has been shown to be beneficial in relieving symptoms of IBS patients with diarrhea[27], highlighting the clinical significance of serotonin-producing EC cells in the development of functional bowel dysfunctions. However, little is known about the regulation of the ISC compartment and its differentiation into EC cells during early-life stress-induced intestinal injury. Here, we discover that NMS expands the ISC compartment and EC cell niche via NGF-mediated TrkA signaling, which is reversed by inhibition of TrkA signaling. Recombinant NGF directly targets ISCs, promoting ISC expansion and its differentiation into EC cells in intestinal organoids by trans-activating Wnt/β-catenin signaling. These findings reveal the regulatory role for NGF in ISC renewal and differentiation during pathological conditions. Understanding how NGF alters intestinal functions will provide insights into ways of reducing the deleterious impact of early-life stress on intestinal integrity.

## Results

**Early-life stress induces enterochromaffin cell hyperplasia.** To investigate the potential role of NGF in early-life stress-induced intestinal changes, we adapted a model of neonatal maternal separation, in which neonatal rodents were individually separated from their mothers for 3 h per day at postnatal 3–14 days and killed for analyses at 2 months upon maternal deprivation[18]. The expression of NGF was first examined in the proximal colon of NMS rats by western blotting analyses. In consistence with previous reports, NMS indeed significantly increased the expression of NGF in the proximal colons of rats (Supplementary Fig. 1a)[19]. The expression of TrkA was similarly upregulated in the proximal colons of NMS rats (Supplementary Fig. 1b)[19]. Immunostaining using a specific antibody against TrkA confirmed enhanced expression of TrkA in the colonic epithelium and crypts upon NMS (Supplementary Fig. 1c). When comparing with the control rats, the NMS rats were more sensitive to visceral pain during the colorectal distention test, as indicated by their lowered intra-luminal pressure (Supplementary Fig. 1d). Rats treated by intra-peritoneal injection of recombinant NGF were found to possess visceral pain thresholds comparable with those of NMS rats (Supplementary Fig. 1d), indicating that either NGF treatment or NMS leads to visceral hypersensitivity. In line with visceral hypersensitivity resulted from NMS/NGF treatment, the serotonin contents of proximal colons from both NMS and NGF-treated rats were remarkably increased when compared with the control rats (Supplementary Fig. 1e). Similar findings were observed in mice subjected to either NMS or NGF treatment (Fig. 1a). The increase in the colonic serotonin level induced by either NMS or NGF treatment persisted for 6 months upon the treatment, indicating that early-life stress causes a long-lasting disturbance in serotonin homeostasis (Supplementary Fig. 2). As EC cells are the primary site for serotonin production, we examined the density of ECs by Masson-Fontana silver staining. Indeed, both NMS and NGF-treated rats exhibited increased density of EC cells in the proximal colons (Supplementary Fig. 1f). To test whether pathophysiological changes in colons resulted from NMS are a consequence of altered NGF signaling, neutralizing antibodies against NGF that block the binding of NGF to its receptor were administered intraperitoneally into NMS rats. Inhibition of NGF signaling in NMS rats completely restored the increased density of the EC cell, the enhanced colonic serotonin content, and the reduced visceral pain threshold, to

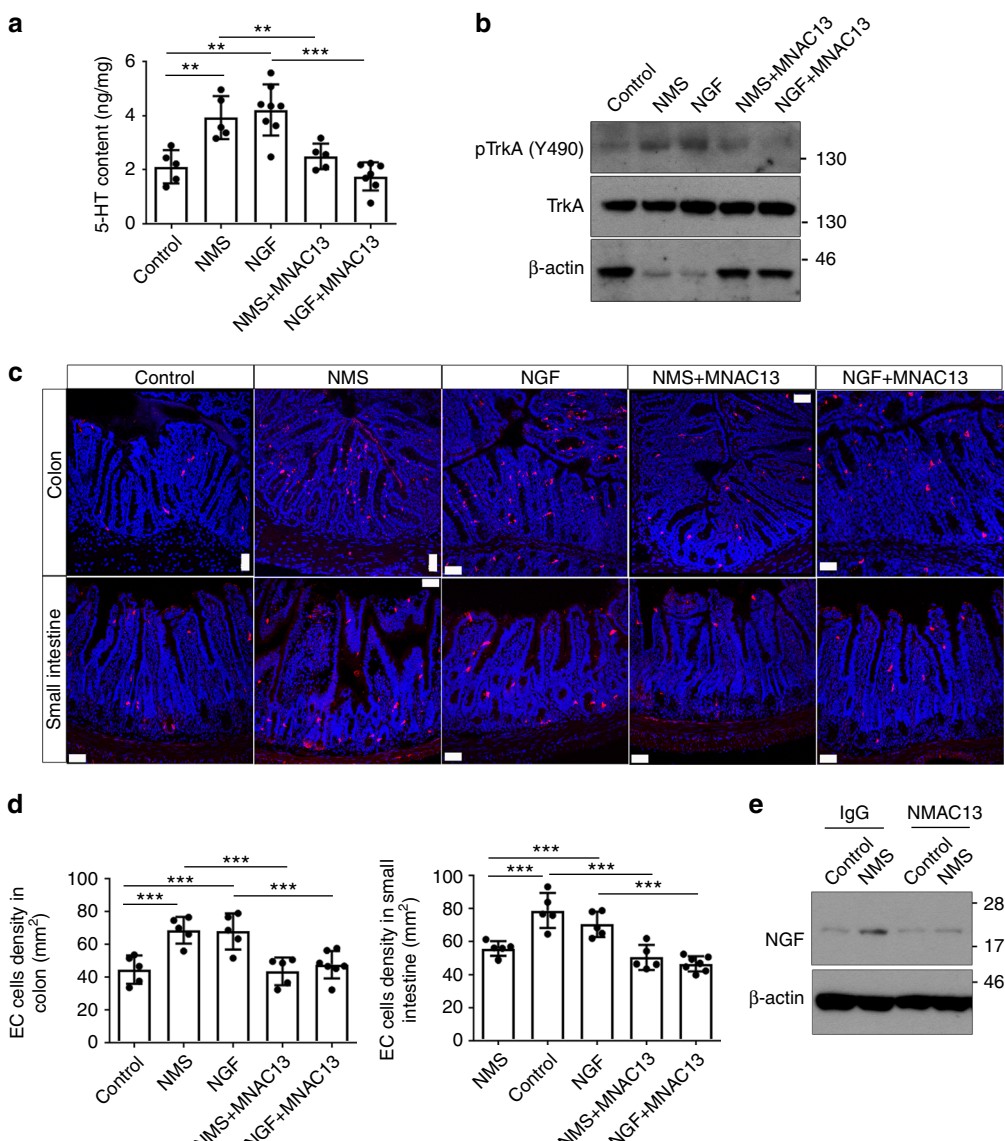

**Fig. 1** NMS leads to EC cell hyperplasia and increased enteric serotonin via NGF/TrkA signaling. **a** The concentration of serotonin/5-HT in the proximal colons isolated from control and NMS mice treated with/without NGF and MNAC13 was measured at 8 weeks upon NMS treatment. **\*\***$p < 0.01$; **\*\*\***$p < 0.001$, $n \geq 5$/group; ANOVA. **b** Western blotting analyses on the expression of p-TrkA (Y490) in the tissues isolated from **a**. The protein loading for total TrkA was normalized and served as a loading control. **c** Immunostaining for serotonin/5-HT shows the distribution of serotonin-producing EC cells in both small intestines and proximal colons from control and NMS mice treated with/without NGF and MNAC13 (scale bar: 50 μm). **d** Quantification for the densities of EC cells in both small intestines (right panel) and colons (left panel). **\*\*\***$p < 0.001$, $n \geq 5$/group, ANOVA. **e** Western blotting analyses on the expression of NGF in the proximal colons from control and NMS mice treated with/without NMAC13 at 8 weeks upon NMS treatment. β-actin served as a loading control. Data represent the mean ± SEM

the levels similar to those observed in the control rats (Supplementary Fig. 1d–f). Though the majority of NGF functions are mediated via the activation of TrkA, NGF is also known to interact with the p75 neurotrophin receptor (p75 NTR) at a much lower affinity[28]. To investigate if NGF/NMS-induced phenotypic changes are mediated via TrkA, NMS/NGF-treated mice were intraperitoneally administered with MNAC13, a well-characterized anti-TrkA monoclonal antibody with remarkable neutralizing properties[29–32]. Indeed, MNAC13 effectively inhibited the NGF-induced phosphorylation of TrkA and Akt in 3T3 cells expressing TrkA, but did not suppress the phosphorylation of TrkB/C induced by BDNF and NT-3 in 3T3 cells expressing TrkB/C (Supplementary Fig. 3). This result further validates the specificity of MNAC13 against TrkA. Consistent with the results obtained from rats, blocking NGF/TrkA signaling in NMS/

NGF-treated mice also effectively reduced the upregulation of phosphorylated TrkA in colons; the increased density of serotonin-producing EC cells in both small intestines and colons, as well as elevated colonic serotonin levels (Fig. 1a–d). Furthermore, inhibiting TrkA did not only rescue the phenotypic changes induced by NMS but also reduced the increased expression of NGF in the colons of NMS mice to the level detected in the colons of control mice (Fig. 1e). These results suggest that early-life stress mediated via NGF/TrkA signaling induces EC hyperplasia and increases the secretion of serotonin in the gastrointestinal tract, triggering visceral hyperalgesia in the animal model.

**Early-life stress alters epithelial secretory lineages.** As NGF/NMS treatment resulted in EC cell hyperplasia, we hypothesized

that NGF/NMS may also affect other intestinal secretory cells. To address this hypothesis, we examined the distribution of Paneth and goblet cells in mice upon NMS/NGF treatment. In line with the increased density of EC cells, we observed a profound increase in the number of Paneth cells in the small intestines isolated from NMS and NGF-treated mice, which was completely reversed by blockade of NGF/TrkA signaling by the treatment with MNAC13 (Supplementary Fig. 4a, b). However, the number of goblet cells was not altered in both small intestines and colons by NMS/NGF treatments (Supplementary Fig. 5a–c). These data suggest that early-life stress alters the specification of secretory lineages in the intestinal epithelium via activation of NGF/TrkA signaling.

**Early-life stress leads to expansion of the ISC compartment.** The intestinal epithelium is daily replenished by intestinal stem cells marked by leucine-rich repeat containing G-protein-coupled receptor 5 (Lgr5), a receptor for Wnt agonists (R-spondins)[33]. Lgr5+ ISCs can differentiate into all cell types in the intestinal epithelium[23]. Given that early-life stress alters the specification of intestinal secretory lineages, we therefore sought to examine whether the NMS affects the homeostasis of ISCs. To investigate the impact of early-life stress on ISC homeostasis, we made use of *Lgr5-EGFP-IRES-CreERT2(lgr5-EGFP)* mice, in which Lgr5+ ISCs are marked with enhanced green fluorescent protein (EGFP)[33]. In both NMS mice and control littermates, immunofluorescent staining for TrkA using sections of proximal colons from *lgr5-EGFP* mice revealed that TrkA colocalized with Lgr5 within the colonic crypt (Fig. 2a). The expression of TrkA in Lgr5+ colon stem cells (CSCs) supports the notion that NGF/TrkA signaling may contribute to the regulation of the gastrointestinal stem cell niche. When compared with the control mice, Lgr5+ CSCs extend toward the upper part of the colonic crypts in both NMS and NGF-treated mice (Fig. 2b, d), indicating the expansion in the CSC compartment and enhanced self-renewal of CSCs in response to NMS/NGF treatment. The expansion in the CSC compartment was further substantiated by the marked increase in the number of Sox9+ cells within colonic crypts, in both NMS and NGF-treated mice (Fig. 2c, e). To investigate whether the expansion in the CSC compartment is dependent on NGF/TrkA signaling, *lgr5-EGFP* mice were intraperitoneally injected with NMAC13 during NMS treatment. Abrogating TrkA signaling by the treatment with NMAC13 indeed abolished the expansion of CSCs in NMS mice (Fig. 2b–e). Similar expansion in Lgr5+Sox9+ISCs was also observed in small intestines from *lgr5-EGFP* mice receiving NMS/ NGF treatment (Supplementary Fig. 6). These results collectively revealed that early-life stress leads to NGF/TrkA signaling-dependent expansion of the ISC compartment in vivo.

To further define the role of NGF in ISCs in the intestinal crypts, we took advantage of an in vitro intestinal organoid culture system to mimic the self-renewal and differentiation of ISCs in the intestinal epithelium. Intestinal crypts were isolated from *lgr5-EGFP* mice and cultured in the standard organoid media containing R-spondin-1, EGF, and Noggin[23]. Surprisingly, addition of recombinant NGF (rNGF) in the culture medium significantly increased the size of organoids and the efficiency of organoid formation at both day 3 and day 6 in culture (Fig. 3a–c), indicating that NGF enhances the self-renewal of ISCs in organoids. Consistent with previous reports, the expression of Lgr5 was restricted to the crypt-like region at the bud of organoids (Fig. 3d)[23]. In contrast, Lgr5-expressing cells were expanded throughout the organoids cultured with NGF (Fig. 3d). The increased expression of Lgr5 in NGF-treated organoids was also detected by both western blotting and quantitative PCR (qPCR) analyses (Fig. 3e, f), suggesting that NGF promotes ISC expansion in vitro. The expansion of ISCs was further supported

by the augmented transcriptional expression of other ISC markers *Sox9* and *Olfm4* in organoids incubated with rNGF (Fig. 3f). To determine whether NGF directly targets ISCs, purified Lgr5-EGFP+ ISCs were isolated by fluorescence-activated cell sorting (FACS) and cultured under standard conditions with or without recombinant NGF. Consistent with the data obtained from crypt-derived organoids, recombinant NGF significantly augmented the size of organoids generated from purified ISCs and expanded the population of Lgr5EGFP+ stem cells throughout the organoids (Supplementary Fig. 7a–c). To investigate if NGF-induced changes in organoids are mediated via TrkA, MNAC13 was applied to the organoid culture. Inhibiting TrkA by MNAC13 effectively suppressed the increased phosphorylation of TrkA induced by NGF and the NGF-stimulated growth in ISC-derived organoids (Supplementary Fig. 7d, e). To examine if our observations obtained from mouse organoids are physiologically relevant to human tissues, we cultured human organoids generated from primary colonic tissues with recombinant human NGF (rhNGF). rhNGF indeed significantly increased the size of human colonic organoids (Supplementary Fig. 8a, b). In line with increased organoid size, the expression of LGR5 was elevated in human organoids cultured with rhNGF (Supplementary Fig. 8c).

Since NGF promotes the specification of secretory lineages in vivo, we examined all cell types of secretory lineages in organoids. Culture with rNGF significantly increased the density of ChgA+ enteroendocrine cells in organoids (Fig. 3g, h). The increase in ChgA+ cells was accompanied with more serotonin-producing EC cells observed in rNGF-treated organoids (Fig. 3i, j). The expansion in serotonin-producing EC cells was also observed in human colonic organoids cultured with rhNGF (Supplementary Fig. 8d, e). Furthermore, immunostaining for a lysozyme that marks Paneth cells showed that there was a significant increase in Paneth cell frequency in the crypt-like region of organoids cultured with rNGF (Fig. 3k, l). The increased differentiation of Paneth cell was also revealed by the upregulation of *Sox9*, a Wnt-responsive gene required for both ISC self-renewal and the specification of Paneth cell, in intestinal tissues isolated from NGF-treated mice and in organoids cultured with NGF (Figs. 2c, 3f, 4f). In contrast, the number of goblet cells in organoids was not altered in response to NGF stimulation (Supplementary Fig. 9). Thus, organoids cultured with rNGF in vitro recapitulate the intestinal phenotypes induced by either early-life stress or NGF treatment in vivo. These findings revealed that NGF expands the ISC compartment and promotes the specification of EC cells and Paneth cells, both in vitro and in vivo.

**NGF/TrkA signaling amplifies Wnt signaling in ISCs.** Wnt/ β-catenin signaling is a critical pathway for the maintenance of the ISC compartment[34], prompting us to postulate that NGF may control ISC homeostasis in early-life stress-induced intestinal injury through regulating Wnt signaling. To address this issue, we first examined the expression of β-catenin, a surrogate for canonical Wnt signaling, in NMS/NGF-treated mice. Immunostaining using specific antibodies against β-catenin showed that the expression of β-catenin was significantly increased in the proximal colons from both NMS and NGF-treated mice (Fig. 4a). Similar findings were observed in the small intestines from NMS/NGF-treated mice (Supplementary Fig. 10). The increased expression of β-catenin was also detected in organoids cultured with rNGF when compared with organoids cultured with standard conditions (Fig. 4c, d). In parallel with the β-catenin upregulation, increased phosphorylation of β-catenin at Y142 and Y654, which promotes the activation of β-catenin complex[35,36], was detected in both the colonic tissues of NMS/NGF-treated mice and the organoids cultured with NGF (Fig. 4b, e). However,

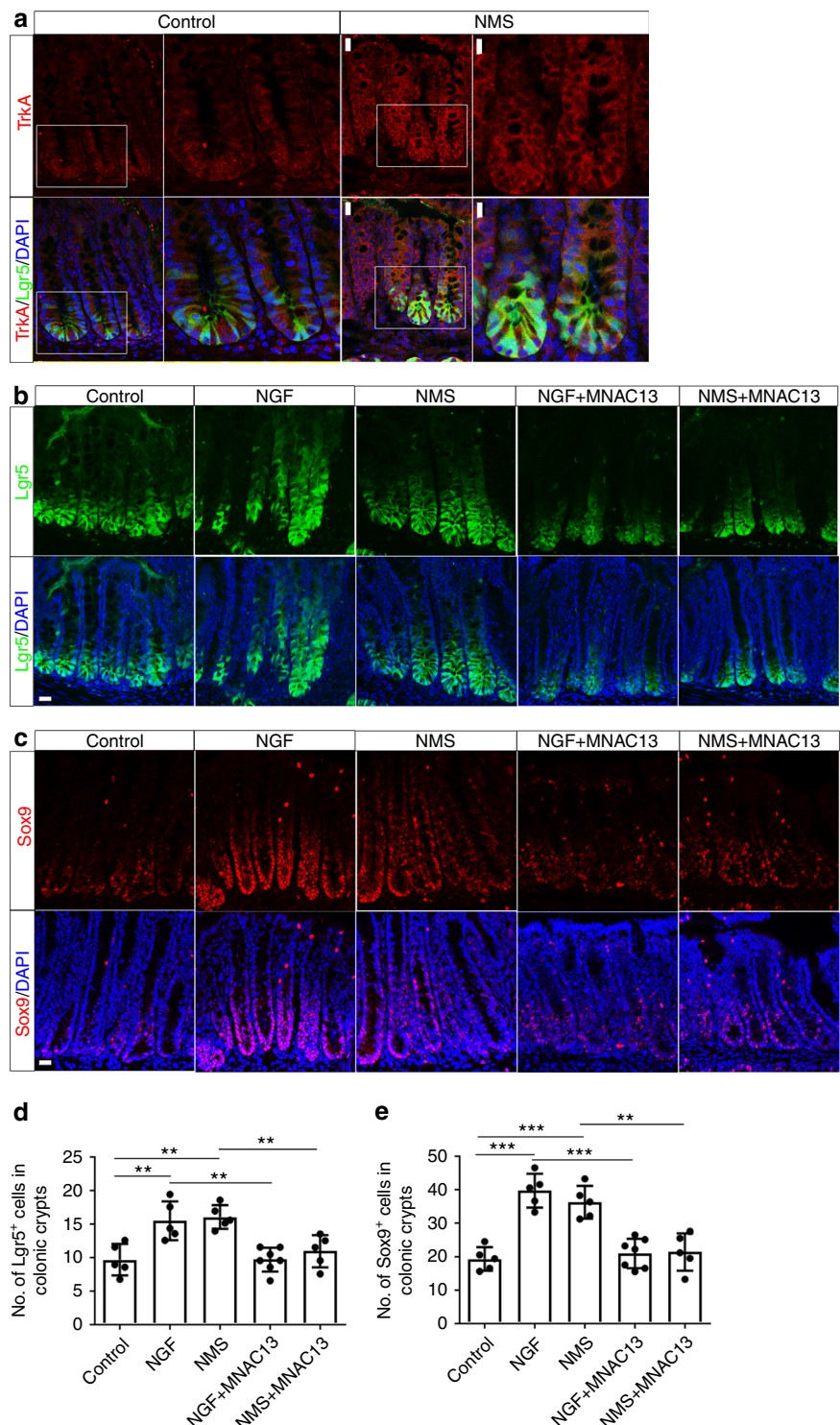

**Fig. 2** NMS results in the expansion in colon stem cell compartment through NGF/TrkA signaling. **a** CSCs from both control (**a–d**) and NMS (**e–h**) Lgr5-EGFP[+] mice were labelled with EGFP (green) in the cross-sections of proximal colons. Immunostaining for TrkA (red) showed that the expression of TrkA localizes to EGFP[+] CSCs. **c**, **d**, **g**, **h** show the higher magnification of white-boxed areas in **a**, **b**, **e**, **f** respectively. (Scale bars: 20 μm for **a, b, e, f**; 10 μm for **c, d, g, h**). **b** CSCs labelled with EGFP (green) in the sections of the proximal colon from both control and NMS-challenged Lgr5-EGFP[+] mice treated with or without intraperitoneal injection of NGF and MNAC13. The number of EGFP[+] cells per colonic crypts were quantified in **d**. **\*\***$p < 0.01$, $n \geq 5$/group; ANOVA (scale bars: 20 μm). **c** Immunostaining for Sox9 (red) in the sections of the proximal colons from both control and NMS Lgr5-EGFP[+] mice treated with or without NGF and MNAC13. **e** shows the quantification for the average number of Sox9[+] cells in colonic crypts. **\*\***$p < 0.01$; **\*\*\***$p < 0.001$, $n \geq 5$/group; ANOVA (scale bars: 20 μm). Nuclei for all panels were visualized by DAPI (blue). All data represent the mean ± SEM

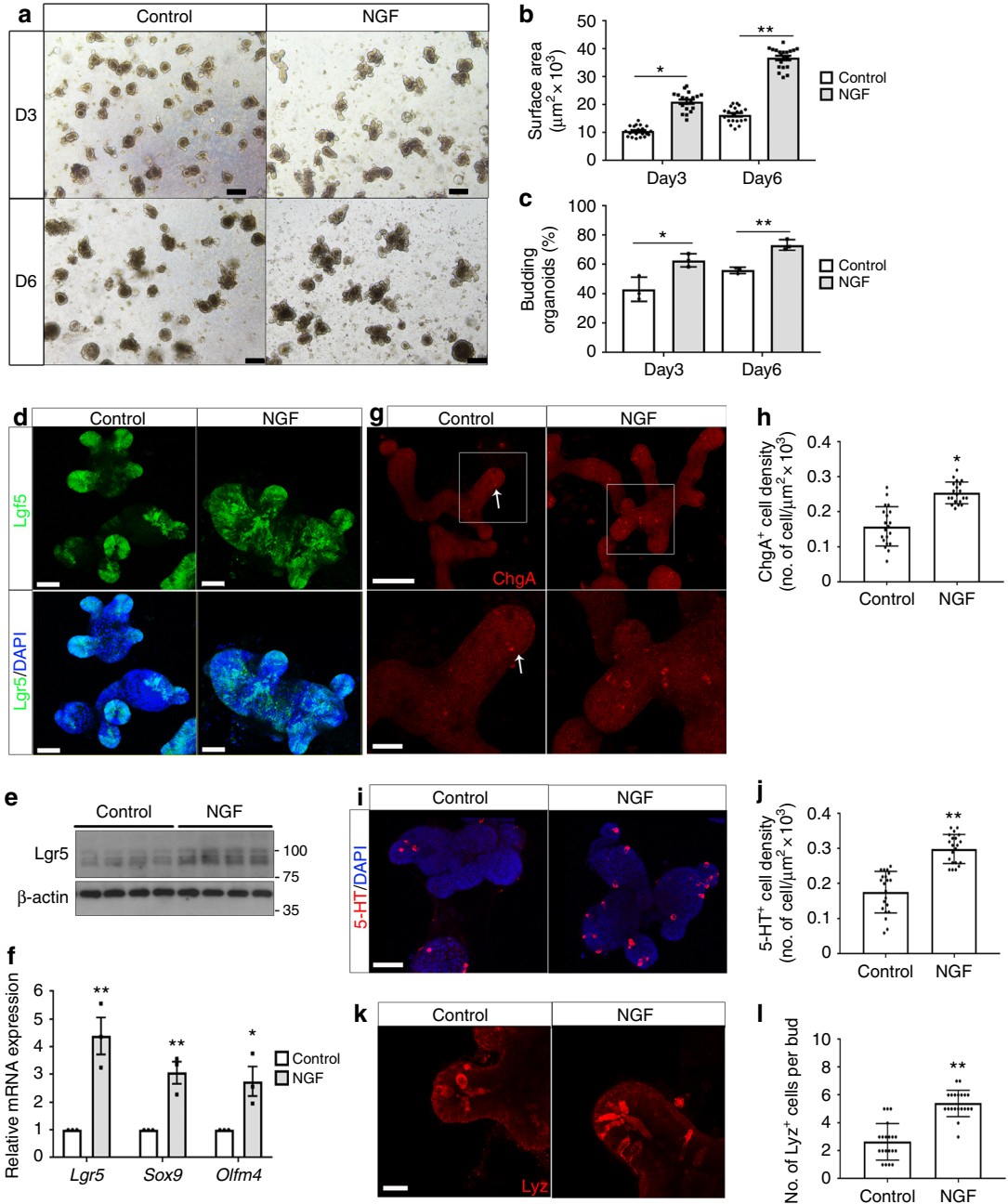

**Fig. 3** NGF enhances the functions of intestinal stem cells in organoid cultures. **a** Representative images show the intestinal organoids cultured with or without recombinant NGF at day 3 (D3) and day 6 (D6) in culture (scale bars: 100 μm). **b** The size of organoids cultured with or without recombinant NGF (*$p < 0.05$, **$p < 0.01$; $n = 20$ organoids per group, two-tailed $t$ test). **c** The efficiency in organoid formation under the culture condition with/without recombinant NGF (*$p < 0.05$, **$p < 0.01$; $n = 3$ per group, two-tailed $t$ test). **d** Whole-mount confocal imaging for the EGFP expression (green) in Lgr5-EGFP+ organoids cultured with/without NGF. Nuclei were counterstained by DAPI (blue) (scale bars: 50 μm for panels **a**, **b**; 100 μm for panels **c**, **d**). **e** Western blotting analyses for Lgr5 expression in organoids with/without NGF. **f** qPCR analyses for the expression of stem cell marker genes (*Lgr5*, *Sox9*, and *Olfm4*) in organoids cultured with/without NGF (*$p < 0.05$, **$p < 0.01$; $n = 3$, two-tailed $t$ test). **g** The enteroendocrine cells in organoids cultured with/without NGF were stained with chromogranin A (Chga) (red) and visualized by whole-mount confocal imaging. The lower panels show the higher magnification of boxed areas in the upper panels. The density for Chga+ in organoids was quantified in **h**. (*$p < 0.05$; $n = 20$ organoids, two-tailed $t$ test) (scale bars: 50 μm for upper panels, 20 μm for lower panels). **i** Whole-mount immunofluorescent co-staining for serotonin/5-HT (red) and DAPI (blue) in organoids cultured with/without NGF (scale bars: 50 μm). **j** Quantification for the density of 5-HT+ EC cells in organoids cultured with/without NGF (**$p < 0.01$; $n = 20$ organoids, two-tailed $t$ test). **k** Confocal imaging shows immunostaining for lysozyme (red) in organoids cultured with/without NGF (scale bars: 20 μm). **l** The Paneth cell frequency per bed was presented by the average number of lyz+ cells per bud in organoids (**$p < 0.01$; $n = 20$ organoids, two-tailed $t$ test). All data represent the mean ± SEM

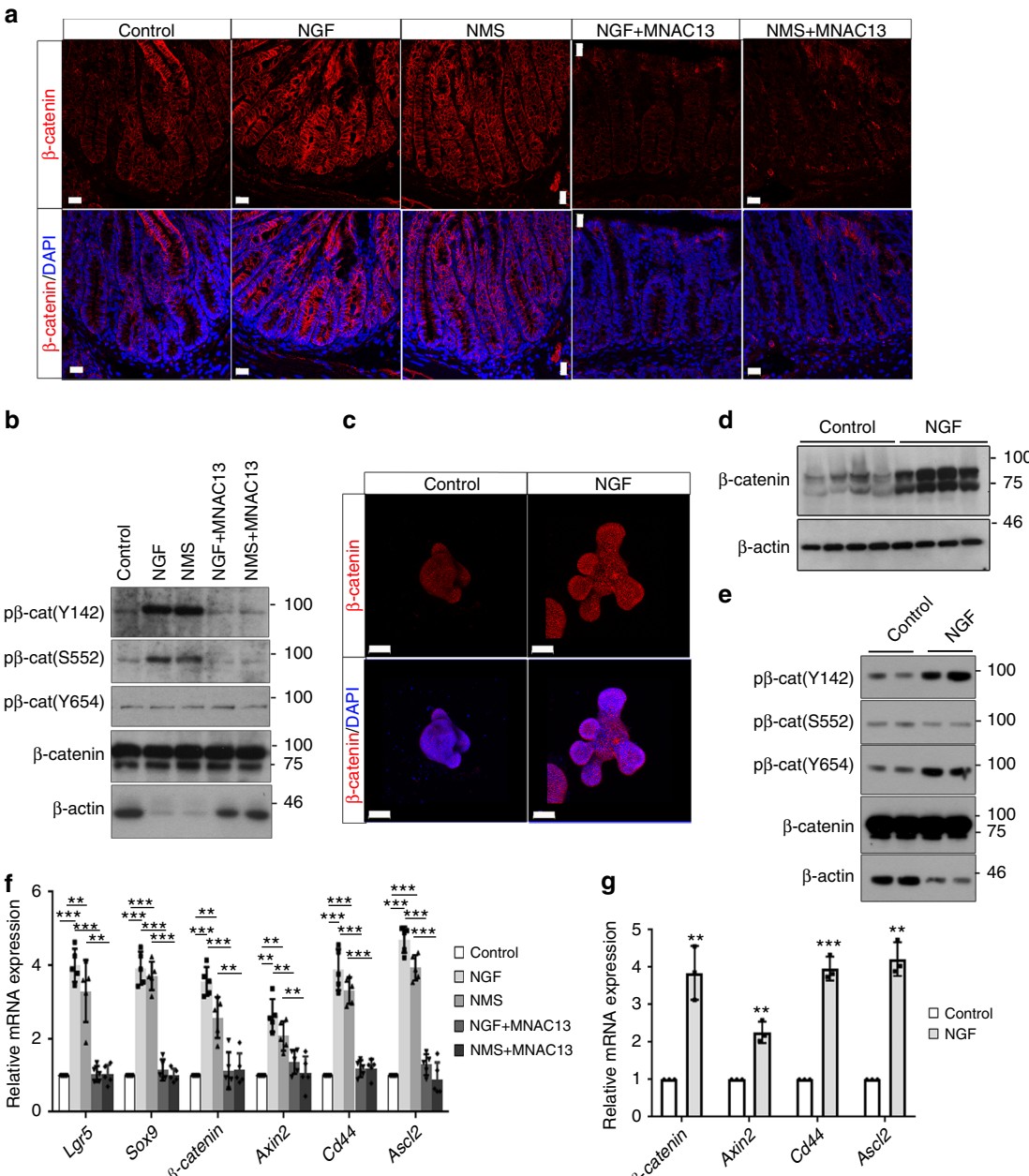

**Fig. 4** Exogenous NGF amplifies Wnt signaling in intestinal stem cells. **a** Immunofluorescent staining for β-catenin (red) in the sections of proximal colons from control and NMS mice treated with or without intraperitoneal injection of NGF and MNAC13 (scale bars: 20 μm). **b** Western blotting analyses on the expression of p-β-catenin (Y142/S552/Y654) in the tissue lysates obtained from **a**. The protein loading for total β-catenin was normalized and served as a loading control. **c** Whole-mount co-staining for β-catenin (red) and DAPI (blue) in organoids cultured with/without NGF (scale bars: 50 μm). **d** Western blotting analyses for the expression of β-catenin in organoids with/without NGF. β-actin served as a loading control. **e** The level of p-β-catenin (Y142/S552/Y654) in organoids cultured with/without NGF was analyzed by western blotting. The protein loading for total β-catenin was normalized and served as a loading control. **f** qPCR analyses on the expression of multiple Wnt-target genes, including *Lgr5*, *Sox9*, *β-catenin*, *Axin2*, *Cd44*, and *Ascl2* in Lgr5EGFP+ cells isolated from control and NMS mice treated with or without intraperitoneal injection of NGF and MNAC13 (**p < 0.01; ***p < 0.001, n ≥ 5/group; ANOVA). **g** qPCR analyses for Wnt-responsive genes, including *β-catenin*, *Axin2*, *Cd44*, and *Ascl2* in organoids cultured with/without NGF (**p < 0.01; ***p < 0.001, n = 3; two-tailed *t* test). All data represent the mean ± SEM

we failed to detect obvious changes in the level of p-β-catenin (S552) (Fig. 4b, e). Inhibiting NGF/TrkA signaling by MNAC13 largely suppressed the NMS/NGF-induced upregulation of both total and phosphorylated forms of β-catenin (Fig. 4a, b). Furthermore, the transcription of *Lgr5* and *Sox9*, two ISC markers regulated by Wnt signaling, was significantly increased in Lgr5EGFP+ intestinal stem cells isolated from NMS/NGF-treated *lgr5-EGFP* mice and in organoids cultured with rNGF (Figs. 3f,

4f). Furthermore, the expression of four other Wnt-responsive genes, such as *β-catenin*, *Axin2*, *Cd44*, and *Ascl2*, was consistently increased in Lgr5EGFP+ cells from NMS/NGF-treated mice and in organoids cultured with NGF (Fig. 4f, g). By the treatment with MNAC13, the increased expression of Wnt-target genes in the Lgr5EGFP+ cells isolated from NMS/NGF-treated mice was reduced to the level comparable with that in cells derived from the control mice (Fig. 4f). These data indicated that NGF/TrkA

signaling amplifies Wnt signaling in vitro and in vivo, probably contributing to the early-life stress-induced intestinal injury.

To further investigate the link between NGF and Wnt signaling pathways, we next tested if NGF activates Wnt/β-catenin signaling by TOP-flash assay, a well-documented luciferase-based reporter assay for Wnt signaling. HEK293 human embryonal kidney cells that were ectopically expressed with both TrkA and Wnt reporters were serum-starved and then stimulated with either NGF or Wnt3a. The administration of either NGF or Wnt3a-activated Wnt reporter activity in a dose-dependent manner, though the NGF-induced reporter activity was generally lower than that induced by Wnt3a (Fig. 5a) when combined, NGF and Wnt3a caused a potent reporter activation exceeding that activated by Wnt3a alone, revealing the synergetic effect between NGF and Wnt3a (Fig. 5g). Similar findings for Wnt activation were observed in Caco2 colon cancer cells that express endogenous TrkA (Fig. 5a). The Wnt reporter activity induced by NGF in Caco2 cells was largely inhibited by the treatment with MNAC13 (Fig. 5b), suggesting that NGF-activated Wnt signaling is mainly mediated by TrkA. In consistence with (Fig. 4b, e), NGF treatment resulted in no obvious changes in the expression of p-β-catenin (S552), but significantly increased the level of p-β-catenin (Y142/Y654) in Caco2 cells (Fig. 5c). The activation of canonical Wnt signaling is initiated by the phosphorylation of low-density lipoprotein receptor-related protein 5/6 (LRP5/6), two cell surface co-receptors for Wnt3a. As previous reports showed that fibroblast growth factor (FGF) signaling-induced Wnt activation is mediated via LRP5/6 phosphorylation[36], we then asked whether NGF-mediated signaling similarly affects the phosphorylation of LRP5/6. Indeed, the phosphorylation of Lrp5 (T1492) and Lrp6 (S1490) in Caco2 cells was significantly increased by NGF stimulation (Fig. 5d). Similar findings were observed in the colonic tissues from NMS/NGF-treated mice (Fig. 5e), indicating that NGF signaling transactivates canonical Wnt signaling by phosphorylating LRP5/6. The NGF-initiated signaling is mainly mediated by the phosphoinositide 3-kinase (PI3K)/Akt pathway and the mitogen-activated protein kinase (MAPK)/Erk1/2 pathway (Fig. 5d)[37]. Chemical inhibition of Erk1/2 by a specific inhibitor U0126 completely abolished the NGF-induced phosphorylation of LRP6, whereas blocking PI3K by Wortmannin had no significant effect on LRP6 phosphorylation (Fig. 5f), indicating that NGF-induced LRP6 phosphorylation is mediated at least by MAPK/Erk1/2 signaling. This finding was further substantiated by the fact that inhibition of Erk1/2, but not PI3K, effectively suppressed NGF-induced Wnt reporter activity in Caco2 cells (Fig. 5g). These findings collectively revealed that NGF/TrkA activates Wnt signaling via Erk1/2 signaling.

To further confirm the role of NGF in regulating Wnt signaling, we examined if NGF can replace R-spondin-1, a Wnt/β-catenin agonist, in the culture for intestinal organoids. Consistent with previous reports, withdrawal of R-spondin-1 eliminated the growth of organoids (Fig. 5h)[38,39]. In contrast, organoids cultured with rNGF in the absence of R-spondin-1 formed with proper 3D structural organization, though their budding efficiency was lower than that of organoids cultured with complete medium (Fig. 5h). Therefore, NGF partially rescues defective organoid formation resulted from loss of Wnt signaling.

**NGF/TrkA regulates ISC expansion through Wnt signaling**. To investigate whether NGF-induced ISC expansion is dependent on Wnt transactivation, we tested if blocking Wnt signaling inhibits the expansion in ISCs induced by NGF/TrkA signaling. The intestinal crypts isolated from *lgr5-EGFP* mice were cultured with a specific Wnt antagonist, IWP-2, that blocks Wnt signaling by inhibiting porcupine (PORCN), a protein important for Wnt

secretion and biological activity. As expected, the budding efficiency of organoids cultured with IWP-2 was dramatically reduced by day 6 when compared with organoids cultured under control conditions with vehicle alone (Fig. 6a, b). In line with reduced organoid formation, the expression of Lgr5 in organoids was almost abrogated by the treatment with IWP-2 (Fig. 6c), suggesting the depletion of ISCs resulted from loss of Wnt signaling. In contrast, organoids cultured with rNGF were relatively resistant to Wnt inhibition by IWP-2, as evidenced by the fact that rNGF partially restored the budding efficiency and the distribution of Lgr5$^+$ ISCs in organoids cultured with IWP-2, to the levels close to but still considerably lower than those observed in organoids cultured under control conditions (Fig. 6a–c). To further verify the crosstalk between NGF and Wnt signaling pathways participating in the ISC expansion in the response to early-life stress, we inhibited Wnt signaling in vivo by daily intraperitoneal administration of IWP-2 into NMS/NGF-treated *lgr5-EGFP* mice. Unlike our previous experimental settings, the distribution of Lgr5$^+$ CSCs in the proximal colon was examined at 16 days of age for mice receiving the NGF/NMS treatment to examine the immediate impact of either maternal deprivation or continuous exposure to exogenous NGF on the stem cell functions. Consistent with our observations in Fig. 2b, the region with Lgr5$^+$ CSCs significantly expanded in the colonic crypts in both NMS and NGF-treated mice compared with the same cell population in control mice. Consistent with the results obtained from in vitro organoid culture, the treatment with IWP-2 depleted Lgr5$^+$ CSCs in the control mice (Fig. 6d, e). Importantly, IWP-2 treatment also attenuated the expansion in the Lgr5$^+$ CSC compartment in both NMS and NGF-treated mice (Fig. 6d, e). Taken together, these results suggested the central role for NGF/TrkA signaling, mediated through Wnt signaling, in the stem cell expansion triggered by early-life stress.

**The correlation between NGF and serotonin in IBS-D patients**. To determine whether our observations obtained from animal models are pathologically relevant to human diseases, we examined the concentrations of NGF and serotonin in the sera derived from IBS-D patients. By means of enzyme-linked immunosorbent assay (ELISA), we found that both NGF and serotonin are markedly upregulated in the sera of IBS-D patients (Fig. 7a, b). Moreover, there was a significant positive correlation between serum NGF and serotonin (Fig. 7c), reinforcing the regulatory role for NGF in serotonin production in early-life stress-associated bowel disorders.

## Discussion

Our study for the first time reports that early-life stress triggers the expansion in the ISC compartment and promotes the ISC differentiation toward the secretory lineage to the result in EC cell hyperplasia, which can be reversed by pharmacological inhibition of NGF/TrkA signaling. Though the majority of NGF functions are mediated via the activation of TrkA, NGF is also known to interact with the p75 neurotrophin receptor (p75 NTR) at a much lower affinity[28]. Given that blockade of NGF/TrkA effectively ameliorates the early-life stress-induced bowel dysfunctions, it appears that the NGF/TrkA axis, but not the NGF/p75 axis, plays a major role in the regulation of intestinal homeostasis under stressful conditions in early life. Many clinical studies have investigated the therapeutic potential of the anti-NGF antibody for pain management in various diseases[40]. TrkA inhibitors have also been tested for their efficacy in the treatment of cancers characterized by the expression of oncogenic Trk fusion proteins[41,42]. This pilot study, together with our prior findings, suggests that the anti-NGF antibody and TrkA inhibitors may

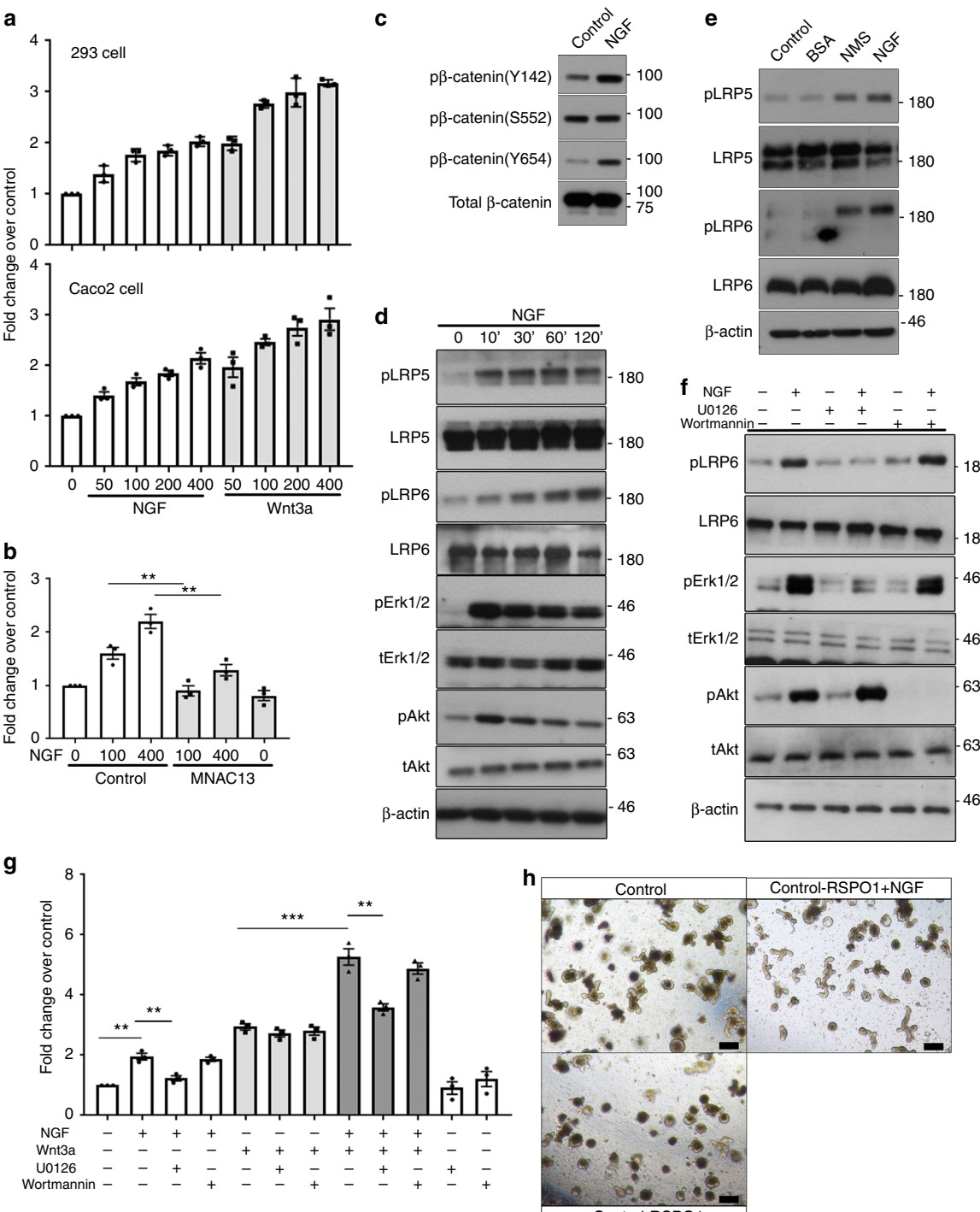

**Fig. 5** NGF/TrkA signaling transactivates Wnt signaling via MAPK signaling. **a** HEK293 cells (upper panel) with ectopic expression of TrkA and Caco2 cells (lower panel) were transfected with the TOPFlash plasmid. Serum-starved cells were treated with either NGF or Wnt3a at indicated concentrations (ng/ml) 24 h prior to the luciferase reporter assay. TCF-binding activities were measured as the readings of the firefly luciferase reporter and were normalized to the reference reporter renilla luciferase. **b** Serum-starved Caco2 cells transfected with the TOPFlash construct were treated with or without MNAC13 prior to NGF stimulation and then subjected to analyses for luciferase activity (**$p < 0.01$, $n = 3$; ANOVA). **c** Western blotting analyses on the expression of p-β-catenin (Y142/S552/Y654) in serum-starved Caco2 cells after NGF treatment. **d** Serum-starved Caco2 was treated with NGF for indicated times. Phosphorylation of Lrp5 (T1492) and Lrp6 (S1490) was detected by western blotting. Total Lrp5/6 served as a loading control. Phosphorylation of Akt and Erk1/2 was a positive control for showing the activation of NGF signaling. **e** Western blotting analyses on the level of phosphorylated forms of Lrp5 (T1492) and Lrp6 (S1490) in the colonic tissues from both control and NMS mice treated with or without intraperitoneal injection of NGF; it is noted that the tissues were isolated for analyses shortly after the completion of NGF/NMS treatment. **f** The phosphorylation of Lrp6 detected by western blotting was examined for the responses of Caco2 cells to the stimulation of NGF. The cells were pre-incubated with or without U0126 and Wortmannin before NGF treatment. **g** Luciferase reporter assay for Wnt signaling in Caco2 cells treated with a combination of NGF, Wnt3a, U0126, and Wortmannin (**$p < 0.01$, ***$p < 0.001$, $n = 3$; ANOVA). **h** Representative images showing the intestinal organoids cultured with/without NGF or R-spondin 1 (scale bars: 100 μm)

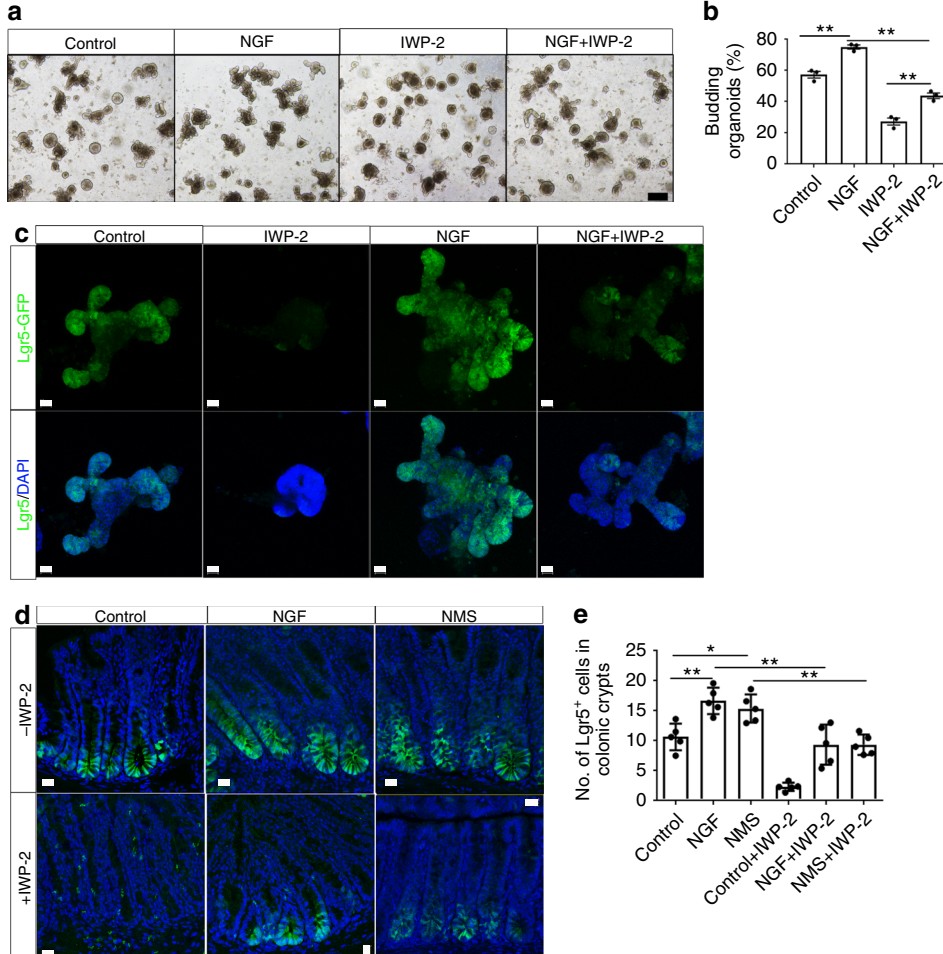

**Fig. 6** NGF promotes the expansion in intestinal stem cells through Wnt signaling. **a** Representative images of intestinal organoids cultured with a combination of NGF and IWP-2 (scale bars: 100 μm). The efficiency for organoid formation was shown in **b** (**$p < 0.01$, $n = 3$ per group; ANOVA). **c** The expression of Lgr5 (green) in Lgr5-EGFP+ organoids cultured in a combination of NGF and IWP-2 was visualized by confocal imaging (scale bars: 20 μm). **d** Confocal imaging shows the distribution of Lgr5+ CSCs (green) in the sections of proximal colon from both control and NMS Lgr5-EGFP+ mice intraperitoneally injected with a combination of NGF and K252a (left panel). The quantification for the average number of Lgr5+ CSCs per colonic crypt was shown in **e** (*$p < 0.05$, **$p < 0.01$, $n = 5$/group; ANOVA) (scale bars: 20 μm). All data represent the mean ± SEM

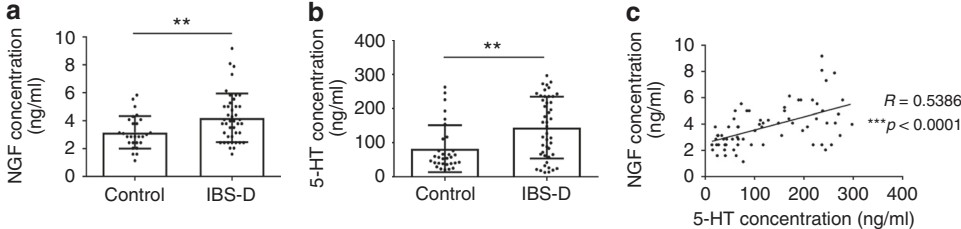

**Fig. 7** NGF in normal intestinal homeostasis and irritable bowel syndrome. **a, b** ELISA assay reveals that there is a significant upregulation of both NGF (**a**) and 5-HT (serotonin) (**b**) in the sera of IBS patients with diarrhea (IBS-D) ($n = 31$ for healthy controls; $n = 45$ for IBS-D patients) (**$p < 0.01$, two-tailed $t$ test). **c** There is also a highly significant positive correlation between NGF and 5-HT in the sera derived from IBS-D patients and healthy controls (**$p < 0.001$, R = 0.5386, $n = 70$, Spearman's correlation)

also be effective for the treatment of early-life stress-associated bowel disorders by targeting NGF/TrkA axis. To examine the therapeutic potential of NGF/TrkA inhibitors, we treated the NMS mice at the age of 10 weeks with NMAC13. Indeed, blockade of TrkA largely but partially rescued the hyperplasia of serotonin-producing EC cells in the proximal colon of NMS mice (Supplementary Fig. 11a, b). Using animal models and intestinal organoids, we showed that the exogenous NGF promotes ISC self-renewal and expansion, suggesting that the treatment with NGF may accelerate intestinal regeneration upon intestinal injuries. Indeed, ectopic expression of NGF in the intestinal epithelium has been shown to improve intestinal recovery, following dextran sulfate sodium-induced colitis in mice[43]. Taken together, these findings suggest that NGF/Trk-A signaling may be a worthwhile therapeutic target in the management of pathophysiological phenotypes in gastrointestinal diseases.

Most of the previous studies regarding NGF/TrkA signaling focus on its neuronal functions. Although the expression of TrkA in non-neuronal tissues has been reported, the functions of NGF/TrkA signaling in non-neuronal biological events are poorly understood. A recent study found that NGF indirectly enhances the stem cell proliferation by promoting innervation[43]. In gastric cancer, NGF overexpression in the gastric epithelium expands enteric nerves that in turn promote the proliferation of epithelial stem cells, in part through acetylcholine/muscarinic receptor-3/ Wnt signaling[43]. In contrast, our study showed that early-life stress induces NGF expression in the gastrointestinal tract and NGF in turn directly promotes the ISC expansion by trans-activating Wnt signaling. The expansion of ISCs observed in NGF-treated mice was recapitulated in vitro in organoids cultured with recombinant NGF, reinforcing our notion that NGF exerts a cell-autonomous effect on ISCs. Therefore, it is conceivable that NGF may regulate the ISC homeostasis in both cell-autonomous and cell nonautonomous manners. As NGF has been shown to alter the cholinergic innervation to the gut[43], we here cannot exclude the possibility that the early-life stress-induced ISC expansion and other intestinal dysfunctions may also be attributable to aberrant innervation resulted from deregulated NGF signaling.

NGF induces ISC hyperproliferation in transgenic mouse models[43] and in intestinal tissues with exogenous NGF (Fig. 2), revealing the role for NGF in pathological settings. However, the function of NGF in the physiological intestinal development remains unknown. Our study found that Lgr5-EGFP mice that received the treatment with K252a, a broad inhibitor for Trks, from postnatal day 3 to day 14 did not exhibit any significant change in the ISC compartment, compared with mice treated with vehicle alone (Supplementary Fig. 12), revealing that inhibiting NGF/TrkA signaling during a sensitive period of development does not affect ISC homeostasis. This is consistent with previous reports that heterozygous loss of either *Ngf* or *TrkA* does not result in obvious histological defects in development, though their deficiency leads to growth retardation and premature lethality due to the loss of sensory and sympathetic neurons[44,45]. Consistently, apart from increased apoptosis in the spleen and skeletal muscle dystrophy, no obvious phenotype in other peripheral tissues is observed in mice expressing transgenic anti-NGF antibodies[46]. Further investigations will be required to examine the long-term effect of deficiency in NGF/TrkA on the intestinal homeostasis by knocking out either *Ngf* or *Trka* specifically in intestines.

Since most of the studies on IBS-associated phenotypes, including visceral hypersensitivity and hyperalgesia, are performed on the colon, our understanding of the intestinal abnormalities caused by IBS remains limited. Our study for the first time showed that the changes in the colon and intestine, such as the expansion in the stem cell population and EC cell hyperplasia, in response to NMS challenge are similar, suggesting that the regulatory mechanisms for the early-life stress-associated bowel diseases, like IBS, are likely conserved in both the colon and small intestine. However, owing to the discrepancies between the NMS model and human IBS, further investigations will be required to confirm the findings obtained from the NMS model in IBS patients in future.

EC cell hyperplasia and deregulated production of serotonin from EC cells have been implicated in gastrointestinal diseases[25,26]. However, little is known about the regulation of EC cell homeostasis during early-life stress-induced intestinal injury. We for the first time reported that NMS, mediated through NGF/TrkA signaling, promotes the differentiation of secretory lineages, resulting in EC cell hyperplasia concomitant with the enhanced secretion of serotonin from gastrointestinal tracts. Moreover,

there is a highly significant correlation between NGF and serotonin in human and they are markedly upregulated in IBS-D patients. It is anticipated that an elevation of enteric serotonin would be a consequence of either an increase in the number of EC cells or the augmented biosynthesis of serotonin in EC cells. To address the latter possibility, the production of serotonin was examined in QGP-1, a human neuroendocrine cell line that can produce and secrete serotonin. Treatment with rNGF in culture failed to alter the serotonin content in QGP-1 cells, which was illustrated by immunofluorescent staining and ELISA analyses for serotonin production (Supplementary Fig. 13a, b). To this end, we speculate that the augmentation of enteric serotonin in the NMS and NGF-treated mice was plausibly owing to the increased EC cell density rather than the augmented synthesis of serotonin.

We here showed that NGF/TrkA signaling regulates ISC self-renewal and expansion through amplifying Wnt signaling, which is consistent with the general assumption that Wnt signaling promotes the proliferation of ISC. However, activation of Wnt signaling does not only enhance the self-renewal of ISC, but also drives the differentiation of ISCs into secretory lineages[47]. This provides an explanation for our observations that NGF promotes both ISC self-renewal and the differentiation of Paneth and EC cells. The fact that NGF does not alter the specification of goblet cell lineage suggests that NGF signaling may not completely recapitulate canonical Wnt signaling in the regulation of ISC differentiation. This idea is further supported by our observations that NGF cannot completely rescue defective organoid formation due to loss of Wnt signaling. Furthermore, it is unlikely that NGF signaling exerts its biological functions without the involvement of other signaling molecules, as signaling pathways usually interact with each other to form sophisticated signaling networks. In addition to the Wnt-dependent mechanism, Wnt-independent yet NGF-dependent mechanisms may also contribute to the NGF-mediated ISC differentiation.

Hyperproliferative ISCs resulted from aberrant activation of Wnt signaling are characteristic hallmarks for colorectal cancer[48]. Our findings that NGF transactivates Wnt signaling in ISCs suggest that prolonged activation of NGF signaling may lead to gastrointestinal cancers. In fact, transgenic overexpression of NGF in the gastric epithelium has been found to initiate and promote gastric tumorigenesis[43]. Furthermore, the expression of NGF was found to be associated with advanced cancer stage in gastric cancers[43], highlighting the significance of NGF signaling in human gastric cancers. Our findings along with other previous studies therefore suggest that chronic exposure to NGF overexpression may be detrimental to health and increases risks for gastrointestinal diseases, such as IBS and intestinal-type cancers. Given these findings, the potential clinical use of NGF may require re-evaluation within the context of a previously unappreciated role for NGF in the regulation of intestinal homeostasis. In summary, we demonstrate how early-life stress, mediated by NGF/TrkA signaling, increases the lifetime risk for functional bowel disorders. Inhibition of NGF may be an important therapeutic strategy for the treatment of gastrointestinal diseases, especially for IBS.

## Methods

**Human subjects**. IBS patients were diagnosed according to Rome III criteria. Forty five IBS patients and 31 healthy controls with mean age of 49 years were recruited from clinics of the School of Chinese Medicine at Hong Kong Baptist University. As serotonin level is usually associated with IBS with diarrhea predominance, we included mainly IBS patients with diarrhea (IBS-D) ($n = 45$) for the experiment. Informed consent was obtained from each patient and healthy control. The study protocol was performed in accordance with the ethical guideline of the Committee on the Use of Human & Animal Subjects in Teaching & Research at Hong Kong Baptist University, and procedures were approved by the Department of Health in accordance with Hong Kong legislation.

Patients will be included if they have all of the following: (1) meeting of diagnostic criteria for IBS (Rome III); (2) age of 18–65 years; (3) IBS symptom severity scale (IBS-SSS) > 75 points at baseline and during the 2-week run-in period; (4) normal colonic evaluation (colonoscopy or barium enema) within 5 years.

Patients will be excluded if they have one or more of the following: (1) pregnancy or breastfeeding; (2) medical history of inflammatory bowel diseases, carbohydrate malabsorption, hormonal disorder, known allergies to food additives, and/or any other serious diseases; (3) having suicidal intention or attempts or aggressive behavior; (4) use of medications known to influence gastrointestinal transit.

Blood samples were collected from healthy subjects and IBS patients in the morning by fasting for 12 h. The serum serotonin level was measured using ELISA (CSB-E08363h, CUSABIO), while NGF concentration in the serum was determined by MILLIPEX Multiplex assay (HADK2MAG-61K, Millipore).

**Animals.** Pregnant female Sprague-Dawley (SD) rats were obtained from the Laboratory Animal Services Centre of The Chinese University of Hong Kong, Hong Kong SAR, China. Lgf5-EGFP-IRES-CreET2 mice on C57BL/6J background were obtained from the Jackson laboratory. All animals and their borne pups were housed in the animal room at Hong Kong Baptist University, kept on a 12-h light/dark cycle with constant ambient temperature, fed with standard laboratory chow, and applied with water ad libitum. Animals of both sexes were used in the experiments. All animal experiments were performed in accordance with the guideline of the Committee on the Use of Human & Animal Subjects in Teaching & Research at Hong Kong Baptist University and procedures were approved by the Department of Health in accordance with Hong Kong legislation.

**Antibodies.** The antibodies used in this study include the following: anti-TrkA (sc-118, Santa Cruz, 1:200 for immunofluorescent staining; 1:2000 for western blotting); anti-NGF (N6655, Sigma, 1:2000 for western blotting); anti-lysozyme (Rb-372-R7, Thermo Scientific, ready-to-use for immunofluorescent staining); anti-Sox9 (AB5535, Millipore, 1:500 for immunofluorescent staining); anti-chromogranin A (sc-13090, Santa Cruz, 1:200 for immunofluorescent staining); anti-mucin2 (sc-15334, Santa Cruz, 1:200 for immunofluorescent staining); anti-serotonin (ab66047, Abcam, 1:200 for immunofluorescent staining); anti-β-catenin (8480, Cell Signaling, 1:200 for immunofluorescent staining; 1:2000 for western blotting); anti-p-β-catenin (Tyr-142) (CP10811, ECM Biosciences, 1:2000 for western blotting); anti-p-β-catenin (S552) (9566, Cell Signaling, 1:2000 for western blotting); anti-p-β-catenin (Tyr-654) (sc-57533, Santa Cruz, 1:1500 for western blotting); anti-Akt (4685, Cell Signaling, 1:3000 for western blotting): anti-pAkt (4060, Cell Signaling, 1:2000 for western blotting); anti-Erk1/2 (4695, Cell Signaling, 1:3000 for western blotting); anti-pErk1/2 (9101, Cell Signaling, 1:2000 for western blotting); anti-LRP5 (5731, Cell Signaling, 1:2000 for western blotting); anti-pLRP5 (ab203306, Abcam, 1:1000 for western blotting); anti-LRP6 (2560, Cell Signaling, 1:2000 for western blotting); anti-pLRP6 (2568, Cell Signaling, 1:1000 for western blotting); anti-β-actin (12262, Cell Signaling, 1:5000 for western blotting); Alexa Fluor 594-conjugated goat anti-rabbit (A-11012, Invitrogen, 1:500); Alexa Fluor 594-conjugated donkey anti-goat antibody (A11058, Invitrogen, 1:500); goat anti-rabbit antibody conjugated with HRP (sc-2030, Santa Cruz, 1:2000); goat anti-mouse antibody conjugated with HRP (sc-2031, Santa Cruz, 1:2000).

**Neonatal maternal separation.** On postnatal days 3–14, pups were removed from their home cage daily for 3 h consecutively (9:00 am to 12:00 pm). After separation, NMS pups were returned to their mothers' cages and left undisturbed, whereas control pups were nursed as usual. All pups were weaned on postnatal day 22. To investigate the involvement of NGF/TrkA signaling in NMS response, pups were daily administered with/without recombinant NGF (1 μg/kg, Thermo Fisher), anti-NGF antibody (1 mg/kg, Sigma), K252a (50 μg/kg, Sigma), NMAC13 (100 μg/kg, Absolute Antibody), or IWP-2 (20 μM, Sigma) by intraperitoneal injection. Injection of control IgG or dimethyl sulfoxide served as a control. For most of the experiments, animals were killed for phenotypic analyses at 8 weeks upon maternal deprivation. To examine the impact of IWP-2-mediated Wnt inhibition on NGF/NMS-induced intestinal changes, the mice were killed for analyses at the 16 days of age. The diagram showing the timeframe for the experiment is shown in (Supplementary Fig. 14). The animal care and sample analysis were not blinded to the group allocation in the animal experiments.

**Organoid culture.** Approximately 20 cm of small intestines were harvested from about 8-week-old mice and flushed with ice-cold phosphate-buffered saline (PBS). The intestinal segment was then opened by the longitudinal incision, and the villi were removed by mechanical scraping. To isolate the crypts, the segment was cut into 2 -mm fragments that were then incubated into Gentle Cell Dissociation Reagent (STEMCELL Technology) with gentle shaking at room temperature for 15 min. After isolation, crypts cells were filtered with a 70-μm cell strainer. The number of crypts were then counted microscopically. For purification of single Lgr5$^{EGFP+}$ intestinal stem cells from crypts, isolated crypts were incubated with Trypsin in PBS (Gibco, 10 mg/ml) supplemented with DNAse (0.8 μg/μl) for 30 min at 37 °C. Subsequently, dissociated cells were filtered with a cell strainer

with 40-μm mesh. GFP-expressing cells were isolated using FACS Aria model (BD Biosciences).

Organoid culture was performed in accordance to the protocol. To facilitate organoid formation, 300 crypts were suspended in the matrix composed of advanced DMEM/F12 medium (Gibco) and growth-factor-reduced Matrigel in a ratio of 1:3. After the gel polymerization, standard medium containing advanced DMEM/F12 medium, 2 mM Glutamax (Invitrogen), 10 mM HEPES (Sigma), 1 mM N-acetyl-cysteine (Sigma), B27 supplement (Invitrogen), N2 supplement (Invitrogen), recombinant murine EGF (50 ng/ml, Invitrogen), recombinant human R-spondin 1 (500 ng/ml, R&D system), and recombinant murine Noggin (50 ng/ml, Preprotech) was added to the organoid culture. For experiments evaluating the effect of NGF on organoid formation, the standard medium was supplemented with/without recombinant murine NGF (100 ng/ml, Sigma) and Wnt inhibitor IWP-2 (0.5 μM, Sigma).

To investigate the effect of NGF on the human colon, colonic tissues were surgically obtained from five independent patients (three with colorectal cancers and two with hemorrhoid) from the Peking University Shenzhen Hospital. To obtain the normal tissue, a distance of more than 5 cm to the tumors was kept. The colonic tissues were washed thoroughly with ice-cold saline, and the underlying muscle layer was removed from the submucosal layer. The tissues were cut into fine pieces that were then incubated into the Gentle Cell Dissociation Reagent with gentle shaking for 30 min at 4 °C. After isolation, crypts cells were filtered with a 70 -μm cell strainer and then suspended in the matrix composed of advanced DMEM/F12 medium (Gibco) and growth-factor-reduced Matrigel in a ratio of 1:1. The cell-matrix mixture was seeded in 48-well plates (1000 single cells per 25 μl of Matrigel per well). The Matrigel was polymerized for 10 min at 37 °C and 250 μl of the culture medium consisting of advanced DMEM/F12 medium, 2 mM Glutamax (Invitrogen), 10 mM HEPES (Sigma), 1mM N-acetyl-cysteine (Sigma), B27 supplement (Invitrogen), N2 supplement (Invitrogen), recombinant human EGF (50 ng/ml, Invitrogen), recombinant human R-spondin 1 (500 ng/ml, R&D System) and recombinant human Noggin (50 ng/ml, R&D System), 10 mM nicotinamide (Sigma), 10 nM SB202190 (Sigma), and 500 nM A83-01 (Sigma) was added. For the experiments of NGF stimulation, the culture medium was supplemented with recombinant human NGF protein (10 ng/ml, Thermo Fisher).

**Cell treatment.** HEK293T cells and Caco2 cells were cultured in the Dulbecco's modified Eagle's medium (DMEM) (Gibco) supplemented with 10% fetal bovine serum (FBS) and penicillin/streptomycin (100 ng/ml). All cell lines were kindly provided by Professor Zhou Zhongjun at the University of Hong Kong. The cells have recently been tested negative for contamination of mycoplasma. For experiments with plasmid transfection, cells at 80–90% confluence were transfected using Lipofectamine 3000 (Invitrogen) in accordance to the manufacturer's recommended instructions. For the NGF induction experiment, cells were serum-starved for 24 h before the stimulation with growth factors, such as recombinant NGF (100 ng/ml, Sigma) and recombinant Wnt3a (100 ng/ml, Millipore).

**Luciferase assays.** A total of $1 \times 10^5$ 293 T or Caco2 cells were seeded in 12-well plates 24 h before transfection. For testing the activation of TCF binding sites, cells were co-transfected with the firefly luciferase-containing TOPFlash plasmid (200 ng), which was developed by Randall Moon's laboratory and the pRL-TK (80 ng). The firefly luciferase activity was measured using the Dual-luciferase reporter assay system (Promega) and normalized to the RL activity. As 293T cells do not express TrkA, they are overexpressed with the high-affinity NGF receptor TrkA by transfecting with the pCMV5 TrkA (Addgene).

**Immunohistochemistry.** For immunohistochemistry staining, paraffin sections with thickness of 7 μm were de-paraffinized and rehydrated with graded ethanol (100% twice, 90%, and 70%, for 3 min each). The endogenous peroxidase activity in the sections was blocked by incubation with 3% $H_2O_2$ in PBS. The slides were then incubated with primary antibodies for 16 h at 4 °C and probed with horseradish peroxidase (HRP)-conjugated secondary antibodies for 1 h at room temperature. Peroxidase activity was visualized by utilizing the Dako LSAB + System-HRP Kit according to the manufacturer's specifications (DakoCytomation), and all sections were counterstained with hematoxylin. Images were captured using Nikon microscope equipped with SPOT advanced software.

To perform immunofluorescent staining, the intestinal tissues were fixed with 4% (wt/vol) paraformaldehyde, embedded in frozen OCT (Sakura Fintek), and cryrosectioned (~10 μm). Cryrosections were permeabilized for 15 min in PBS + 0.1%Triton (PBST) and blocked for 1 h with 1% bovine serum albumin (BSA) in PBST. The tissues were incubated with primary antibodies overnight at 4 °C, followed by incubation with secondary antibodies at room temperature for 1 h. Immunostained positive signals were detected with a Confocal Laser Scanning Microscope (Leica TCS SP8).

To analyze the organoids by whole-mount immunofluorescent staining, the intestinal organoids cultured in eight-well chamber slides were fixed with 4% (wt/vol) paraformaldehyde, permeabilized for 30 min in PBS + 0.3%Triton, and blocked for 1 h with 1% BSA in PBST. The organoids were then incubated with primary antibodies overnight at 4 °C, followed by incubation with secondary antibodies at room temperature for 2 h. The immunofluorescent images were

captured by Confocal Laser Scanning Microscope (Leica TCS SP8) that was operated with a software, LAS X. A series of z-stacks images were captured within the z-dimensions (around 10–15 layers) to generate three-dimensional microscopic images that were displayed as maximum projection images.

**Western blotting**. Total protein was either extracted from cells or tissues using ice-cold RIPA buffer (25 mM Tris-HCl, 150 mM NaCl, 1% NP-40, 1% sodium deoxycholate, 0.1% SDS, complete protease cocktail (Roche)). Protein samples at 10 μg were separated by 7.5–15% SDS-PAGE and electrophoretically transferred to polyvinylidene difluoride membranes (Bio-rad). After blocking with 5% non-fat milk, membranes were incubated with primary antibodies overnight at 4 °C. After rinsing, membranes were incubated with appropriate secondary antibody conjugated with HRP at room temperature for 1 h. The positive immunoreactions were detected with x-ray film (Fuji) by chemiluminescence using an ECL kit (GE Healthcare). The relative expression of proteins was quantified using Image J software (Wayne Rasband, NIH, USA). The uncropped and unprocessed scans of all blots are available in (Supplementary Fig. 15).

**Real-time quantitative polymerase chain reaction (qPCR)**. At the time of harvest, the total RNA was extracted from either cells or organoids using TRIzol reagent (Invitrogen) according to the manufacturer's instruction. In total, 2 μg of the total RNA of each sample was reversely transcribed into cDNA using Prime-Script RT master mix (Takara) in a total volume of 20 μl. cDNA templates were then amplified with specific primers for target genes in the ABI ViiA 7 real-time PCR system (Applied Biosystems) using 2X SYBR Green PCR Master Mix (Applied Biosystems). Expression of gene of interest of each sample was normalized to the endogenous control GAPDH, and presented as $2^{-\Delta\Delta Ct}$ using the comparative Ct method. Primer sequences for qPCR analyses are listed in Supplementary Table 1.

**Measurement of serotonin concentration in colonic tissues**. Serotonin measurement was carried out on a P/ACE MDQ capillary electrophoresis system equipped with a laser-induced fluorescence detector (Beckman Coulter Instrument). The data were collected and processed by Beckman P/ACE 32 Karat software Version 7.0. The colonic serotonin concentration was expressed as ng/mg (wet weight of tissue).

**Assessment of colonic visceral sensitivity**. Colonic visceral sensitivity of adult rats was measured by means of abdominal withdrawal reflex (AWR) tests. When the rats were slightly sedated with ether, a distension balloon was inserted into the colorectum with the distal tip 1 cm from the anus and secured in place by taping the balloon catheter to the base of the tail. The animal recovered from ether were placed in plexiglass cages ($20 \times 15 \times 15$ cm$^3$) and allowed to adapt for 30 min. Subsequently, the behavioral response to colorectal distention was assessed according to the previous report. The AWR response was tested repeatedly for five times in each rat in 5 min intervals and was evaluated by a blinded observer.

**Enterochromaffin cell staining**. EC cell staining was performed following the procedure of Masson-Fontana silver staining with slight modifications. Briefly, the deparaffined and rehydrated sections were incubated with 5% ammoniacal silver solution in a dark humidified chamber for 4 h at room temperature, followed by a 2-h incubation at 56 °C and an overnight incubation at room temperature. Slides were visualized utilizing the Zeiss Axiovert 200 microscope (Zeiss GmbH). The appearance of brown to black silver precipitate in cytoplasm of EC cells was counted as a positive reaction.

**Periodic acid Schiff (PAS) staining for goblet cells**. The PAS staining was performed using the PAS staining kit (Millipore) in accordance to the manufacturer's suggested protocol. Briefly, the deparaffined sections were hydrated with distilled water. After rehydradation, the sections were incubated with Periodic acid solution (provided) for 5 min and then immersed with Schiff's Solution for 15 min, followed by counterstaining with hematoxylin for 2–3 min. The images were captured with the Zeiss Axiovert 200 microscope (Zeiss GmbH).

**Statistical analyses**. Each experiment was independently repeated for at least three times. Animal experiments involved at least three independent and randomly chosen mice at comparable developmental stages and none of the samples were excluded from analyses. Sample size was determined from the power of the statistical test performed and was increased in accordance to the statistical variation. The statistical differences were determined using one-way analysis of variance (ANOVA) followed by Tukey's post hoc test, Mann–Whitney $U$ test or student's $t$ test. All values are expressed as means ± s.e.m. along with the number of individual mice/samples analyzed ($n$). All data meet the normal distribution. $P$-value of < 0.05 is accepted as statistically significant.

**Reporting summary**. Further information on research design is available in the Nature Research Reporting Summary linked to this article.

## Data availability

The data that support the findings of this study are available within the paper and the supplementary information files or from the corresponding author upon reasonable request.

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

## Acknowledgements

This work was gratefully supported by RGC (260010), HKBU Faculty Research Grant (FRG1/16-17/021), Innovation and Technology Fund (ITS/148/14FP), the National Science Foundation of China (81400596), and Scientific Research Starting Foundation for the Returned Overseas Chinese Scholars from Ministry of Human Resources and Social Security of China. We thank Prof. Zhou Zhongjun for providing us with the *Lgr5-EGFP-IRES-CreERT2* mice.

## Author contributions

H.L.X.W. and H.-Y.Q. designed and conducted most of the experiments. S.W.T. contributed to the experimental design and discussion. X.Z., S.C. and C.F.W.C. helped to collect samples and performed some experiments. L.Z., C.Y.L. H.Y.K. and A.L. provided suggestions for the experimental design. X.L., H.-T.X., T.Y. and F.M.L. provided the materials for the experiments. T.H. contributed to the statistical analyses of the results. H.L.X.W. and Z.-X.B. wrote the paper. Z.-X.B. supervised all aspects of the work.

## Additional information

**Competing interests:** The authors declare no competing interests.

