## [Peer Review File · Nature Communications]

Reviewers' comments:

Reviewer #1 (Remarks to the Author):

Chronic stress during early life can lead to irritable bowel syndrome (IBS), the underlying mechanism remains poorly understood. In this work, Dr. Bian and colleagues have conducted a series of experiments in the mouse and rat models in which pups were periodically separated from mother and showed that elevated NGF is responsible for the IBS phenotype. They further demonstrated that using an inhibitor, K252a, that inhibits NGF/TrkA signaling can rescue the

Phenotype. A key cellular target of NGF was found to be ISCs. As Wnt signaling is the key signaling for ISC self-renewal and proliferation, the authors also revealed that it was TrkA mediated Erk signaling activates Wnt/ β -catenin via phosphorylation of the upstream of Wnt signaling Lrp6. This work not only uncovers the cellular and molecular mechanism underlying IBS, but also provides an insight for clinical treatment of IBS.

Concerns.

1. To confirm that β -catenin is indeed subject to Erk-Lrp6 signaling regulation, the authors should examine whether the increased β -catenin is the form of phospho- β -catenin^{Ter142}, which was reported to be specifically regulated by p-Lrp6 (Krejci Plus One 2012).
2. Which form of p-Lrp6 (Ser1490 or Thr1572) was used in the western blot in Fig5e.
3. The western blot result using Caco2 cells should be verified in intestine in vivo.
4. Please also check the form of Akt phosphorylated form of β -catenin, phospho- β -catenin^{S552}.

Reviewer #2 (Remarks to the Author):

This is a very interesting paper and I find their findings with regard to the impact of maternal separation (MS) and nerve growth factor (NGF) on the expansion of enteroendocrine (EC) and Paneth cells convincing. They have nicely teased out the pathway(s) whereby NGF mediates this expansion and along the way uncover some novel effects of NGF on intestinal stem cell biology.

My concerns relate more to the interpretation of the findings rather than the performance and analysis of the experiments themselves.

1. Yes, MS is a model which recapitulates some aspects of IBS (and visceral hyperalgesia, in particular), but it has many other features including anxiety and depression and behavioral changes (Moloney RD, et al. *Front Psychiatry* 2015;6:1-30). Thus, as acknowledged by the authors, there may other mechanisms whereby MS leads to IBS-type features later in life. Furthermore, no animal model recapitulates the full human phenotype. I would encourage less emphasis, therefore, on the significance of these findings for IBS and more on the biological effects of NGF on the intestinal epithelium.

2. The authors rightly emphasize the proliferative effects of NGF (and MS) but fail to acknowledge that these aspects are not consistent features of IBS in man:

(a) IBS is not consistently associated with mast cell, EC

and certainly not Paneth cell expansion. Indeed, the mast cell story in IBS has proven to be more complex than initially thought (e.g Braak B, et al. *Am J Gastroenterol* 2012;107:715-26) and EC cell expansion has been linked more to post-infection IBS (Dunlop SP, et al. *Gastroenterology* 2003;125:1651-9).

(b) The expansion of Paneth cells is interesting given current interest in the microbiota-gut-brain axis in IBS (in both experimental models, such as MS, and in man) but I am unaware of a description of Paneth cell expansion in IBS.

(c) The hyper-proliferative and potentially carcinogenic effects of NGF make sense but there is no evidence of an association between IBS and colon cancer.

(d) The focus of this study is on the epithelium whose involvement in IBS, in man is unclear, as certainly less well understood than the roles of the enteric nervous and central nervous systems.

3. For various reasons the tissues employed for the various experiments differ - some proximal colon, some small intestine. While the precise locus (or loci) of peripheral abnormalities in IBS are unclear, most studies on visceral hypersensitivity and hyperalgesia (for logistical and access reasons) have been performed on the colon. Findings in the small intestine may not always be applicable to the colon and this should be acknowledged. In man, cell populations, such as those of mast cells, vary considerably along the colon. Why were the immunohistochemical (and related) studies performed in the proximal colon when the distensions were performed peripherally?

4. In the manuscript the authors refer to "early-life stress-induced intestinal injury". What is the evidence that early life stress actually damages the intestine in either animal models or in man? The relationships between early life stress and subsequent physical or psychological distress in later life are complex and somewhat contentious (e.g. Romans S, Cohen M. *Harv Rev Psychiatry* 2008;16:35-54) but I am not aware of any intestinal pathology that is demonstrable. My understanding is that early life stress imposes a biological imprint/memory (in the CNS, ENS, etc.) which makes the individual susceptible to other environmental or internal stimuli later in life.

5. The comments about the therapeutic potential of anti-NGF antibodies and TrkA inhibitors are speculative; will their administration in adult life have any impact? Must they not be given much closer to the time of exposure? The experiments in this paper, as I understand them were performed no later than 8 weeks.

6. I find the human data of little relevance and would exclude it. There are too many leaps of faith from these results and the data from the animal and in vitro work to make them relevant.

7. Other points - are the doses of NGF used pharmacological or biological?

Page 5, line 12 - this sentence is incomplete

Page 5, para 3 - what is the evidence that early-life stress "alters the specification of intestinal cell lineages"?

Reviewer #3 (Remarks to the Author):

This manuscript describes an interesting and novel idea; that early life stress (modeled by maternal separation) can cause later symptoms similar to IBS by working through the NGF-trkA axis, and that this then trans activates the Wnt signaling cascade to modify stem cells. Unfortunately this work suffers from two main deficiencies: the first is that the experiments are described in insufficient detail to actually allow an understanding of what might actually be going on in the system, and second and more importantly, they use reagents of insufficient specificity to actually back up the claims.

To discuss the second point first, the paper relies heavily on the use of K252a as an inhibitor of trkA. K252a does inhibit trkA, but it is not specific, nor even very selective, inhibiting a wide variety of tyrosine kinases and even serine-threonine kinases. This has been known for quite some time (Ruegg and Burgess, *Trends in Pharmaceutical Science*, 10, 218-220, 1989; Mizuno, et al., *Federation of European Biochemical Societies*, 330, 114-116, 1993; Martin et al., *Neuropharmacology* 61, 148-155, 2011 as examples). Use of such a non-selective inhibitor, especially with no information given about the in vivo dose used, the dosing schedule or the timing of the experiments, much less the actual exposure in vivo with assays of drug levels, can tell almost nothing about the proposed involvement of the NGF-trkA system.

Second, throughout the paper, there is insufficient methodological detail to understand what the authors are proposing. They show that neonatal separation can result in a variety of effects much later in life and then state that NGF administration can mimic these. But they never address how much NGF is administered, when it is administered, how often or for how long it is administered or how long after administration the various assays take place. The same is true about the experiments with administration of purported NGF inhibitors, when are they administered, at what dose, how often, for how long and how soon before assays take place? Without this information, it is not possible to try to understand what might be going on in the animal. Does neonatal separation induce an NGF and/or trkA upregulation that lasts for life and needs to be acutely and continuously inhibited for cessation of the effects? Or is inhibition some sort of “breaking the cycle” that leads to a long term cessation? And even more basically, what is the effective dose of the non-selective inhibitor K252a?

This system is of obvious interest but there are other issues with the work in this paper (the in vitro dose of NGF used is exceptionally high and calls into question the relevance of any effects seen). Overall, the fact that the interpretations in this work depend so heavily on K252a and that the experiments are insufficiently described makes it impossible to recommend this paper for publication.

Reviewer #1:

Chronic stress during early life can lead to irritable bowel syndrome (IBS), the underlying mechanism remains poorly understood. In this work, Dr. Bian and colleagues have conducted a series of experiments in the mouse and rat models in which pups were periodically separated from mother and showed that elevated NGF is responsible for the IBS phenotype. They further demonstrated that using an inhibitor, K252a, that inhibits NGF/Trka signaling can rescue the phenotype. A key cellular target of NGF was found to be ISCs. As Wnt signaling is the key signaling for ISC self-renewal and proliferation, the authors also revealed that it was Trka mediated Erk signaling activates Wnt/b-catenin via phosphorylation of the upstream of Wnt signaling Lrp6. This work not only uncovers the cellular and molecular mechanism underlying IBS, but also provides an insight for clinical treatment of IBS.

We thank the reviewer for the positive comments on our study. We also highly appreciate the insightful critics for improving our study.

Concerns.

1. To confirm that b-catenin is indeed subject to Erk-Lrp6 signaling regulation, the authors should examine whether the increased b-catenin is the form of phosphorylated b-catenin at Ser142, which was reported to be specifically regulated by p-Lrp6 (Krejci Plus One 2012).

***Answer:** Thank you for the suggestion. We detected increased expression of p- β -catenin (Ter142) in response to NGF stimulation in both Caco2 cells (**revised Fig. 5c**) and intestinal organoids (**revised Fig. 4e**) by western blotting. Moreover, elevated phosphorylation of β -catenin at Ter142 residue was also observed in the tissue lysates of proximal colons from both NMS and NGF-treated mice (**revised Fig. 4b**).*

2. Which form of p-Lrp6 (Ser1490 or Thr1572) was used in the western blot in Fig 5e. ***Answer:** p-Lrp5(T1492) and p-Lrp6 (S1490) were detected in the western blot. We have added the information in the revised figure legend for Fig. 5.*

3. The western blot result using Caco2 cells should be verified in intestine in vivo. ***Answer:** In line with the results obtained from Caco2 cells, we detected increased expression of p-Lrp5(T1492) and p-Lrp6 (S1490) in the small intestines isolated from NMS and NGF-treated mice by western blotting (**revised Fig. 5e**).*

4. Please also check the form of Akt phosphorylated form of b-catenin,

phosphor-b-cateninS552.

Answer: *Following the reviewer's suggestion, we analyzed the expression of p- β -catenin (S552/Y654). Western blotting analyses showed increased phosphorylation of β -catenin at Y654 in response to NGF stimulation in both Caco2 cells (**revised Fig. 5c**) and intestinal organoids (**revised Fig. 4e**). However, there was no significant change in the expression of p- β -catenin (S552) upon the NGF challenge (**revised Fig 4e & Fig. 5c**). Similar results were observed in the proximal colons from both NMS and NGF-treated mice (**revised Fig. 4b**). Overall, we found that NGF stimulates the phosphorylation of β -catenin at both Y142 and Y654 in the gut in vitro and in vivo.*

Reviewer #2:

This is a very interesting paper and I find their findings with regard to the impact of maternal separation (MS) and nerve growth factor (NGF) on the expansion of enteroendocrine (EC) and Paneth cells convincing. They have nicely teased out the pathway(s) whereby NGF mediates this expansion and along the way uncover some novel effects of NGF on intestinal stem cell biology.

My concerns relate more to the interpretation of the findings rather than the performance and analysis of the experiments themselves.

We thank the reviewer for the encouraging comments and the constructive criticism that helped us improve the manuscript.

1. Yes, MS is a model which recapitulates some aspects of IBS (and visceral hyperalgesia, in particular), but it has many other features including anxiety and depression and behavioral changes (Moloney RD, et al. Front Psychiatry 2015;6:1-30). Thus, as acknowledged by the authors, there may other mechanisms whereby MS leads to IBS-type features later in life. Furthermore, no animal model recapitulates the full human phenotype. I would encourage less emphasis, therefore, on the significance of these findings for IBS and more on the biological effects of NGF on the intestinal epithelium.

Answer: *To the best of our knowledge, NMS model should be the best animal model for IBS though it does not fully recapitulate the human phenotypes. Due to the technical limitation, we have followed the reviewer's advice and revised the manuscript by reducing our emphasis on IBS (pls. see P.2 lane 2-6; P.3 lane 5-9)*

2. The authors rightly emphasize the proliferative effects of NGF (and MS) but fail to acknowledge that these aspects are not consistent features of IBS in man:

Answer: *Despite the acknowledged discrepancies between the human IBS and animal model, NMS model should be the best animal model that provides an alternative to costly and more ethically investigate human IBS. Following the reviewer's suggestions, we revised the manuscript by putting less emphasis on IBS. We also acknowledged this issue in the revised Introduction. (pls. see P.2 lane 2-6; P.3 lane 5-9)*

(a) IBS is not consistently associated with mast cell, EC and certainly not Paneth cell expansion. Indeed, the mast cell story in IBS has proven to be more complex than initially thought (e.g Braak B, et al. Am J Gastroenterol 2012;107:715-26) and EC cell expansion has been linked more to post-infection IBS (Dunlop SP, et al. Gastroenterology 2003;125:1651-9).

Answer: *In addition to post-infection IBS, the expansion in serotonin-producing EC cells was also reported in the colonic biopsy of IBS-D patients¹. This research group also reported the positive correlation among the number of colonic EC cells, the amount of mucosal serotonin release and mast cell counts¹.*

(b) The expansion of Paneth cells is interesting given current interest in the microbiota-gut-brain axis in IBS (in both experimental models, such as MS, and in man) but I am unaware of a description of Paneth cell expansion in IBS.

Answer: *Due to the technical limitation, it is difficult to obtain intestinal biopsy from both IBS patients and healthy controls, which explains why our understanding on Paneth cells in IBS remains limited. We agree with the reviewer that it will be interesting to investigate the changes in Paneth cells in human IBS.*

(c) The hyper-proliferative and potentially carcinogenic effects of NGF make sense but there is no evidence of an association between IBS and colon cancer.

Answer: *We are sorry that our statements in the discussion are not clear. We do not propose the association between IBS and colon cancer in this study. Our study just suggests that chronic exposure to high level of NGF may increase risks and susceptibility for various gastrointestinal diseases, such as IBS and GI cancers. It however does not necessarily indicate that IBS is associated with GI cancers. We will remove the part of cancer in the discussion if the editor and other reviewers agree to do so.*

(d) The focus of this study is on the epithelium whose involvement in IBS, in man is unclear, as certainly less well understood than the roles of the enteric nervous and central nervous systems.

Answer: *We agree with the reviewer that the role of epithelium in human IBS is not well-explored, which is the initiative goal for this study.*

3. For various reasons the tissues employed for the various experiments differ - some proximal colon, some small intestine. While the precise locus (or loci) of peripheral abnormalities in IBS are unclear, most studies on visceral hypersensitivity and hyperalgesia (for logistical and access reasons) have been performed on the colon. Findings in the small intestine may not always be applicable to the colon and this should be acknowledged.

Answer: *Most of our key animal experiments employ both colon and small intestine and the results obtained from both tissues consistently show the expansion in stem cell population and EC cell hyperplasia in response to NMS and NGF treatment, indicating that the regulatory mechanism for NMS/NGF-induced GI changes are likely conserved for both colon and small intestine. We agree with the reviewer that the intestinal abnormalities caused by IBS in human remains unclear. Our results obtained from murine small intestine will provide hints revealing potential intestinal changes in human IBS, which will hopefully be beneficial for IBS research in the future. Following the reviewer's suggestion, we addressed this issue in the discussion.*

(pls. see P.10 lane 30-36)

In man, cell populations, such as those of mast cells, vary considerably along the colon. Why were the immunohistochemical (and related) studies performed in the proximal colon when the distensions were performed peripherally?

Answer: *We examined the changes in EC cell density in both distal and proximal colons after NMS and found that the changes in EC cells are more dramatic in proximal colons. Therefore, we performed most of the experiments using proximal colons.*

4. In the manuscript the authors refer to "early-life stress-induced intestinal injury". What is the evidence that early life stress actually damages the intestine in either animal models or in man? The relationships between early life stress and subsequent physical or psychological distress in later life are complex and somewhat contentious (e.g. Romans S, Cohen M. Harv Rev Psychiatry 2008;16:35-54) but I am not aware of any intestinal pathology that is demonstrable. My understanding is that early life stress imposes a biological imprint/memory (in the CNS, ENS, etc.) which makes the individual susceptible to other environmental or internal stimuli later in life.

Answer: *We are sorry for our inappropriate wordings. We have rephrased "early life stress induced intestinal injuries" into "the early life stress induced intestinal changes"*

in the revised manuscript.

5. The comments about the therapeutic potential of anti-NGF antibodies and TrkA inhibitors are speculative; will their administration in adult life have any impact? Must they not be given much closer to the time of exposure? The experiments in this paper, as I understand them were performed no later than 8 weeks.

Answer: *Following the reviewer's suggestion, we treated NMS-treated mice with the anti-TrkA antibodies (NMAC13) in the adult life (8 weeks upon NMS) for 10 days. We found that blockade of TrkA effectively reduces the number of serotonin producing EC cells in the proximal colons of NMS mice although the EC cell number in the NMAC13-treated NMS mice is still considerably more than that of control mice (revised supplementary Fig. 9a-b). These findings reveal the therapeutic potential of TrkA inhibitor in the treatment of early life stress-associated bowel disorders, such as IBS.*

6. I find the human data of little relevance and would exclude it. There are too many leaps of faith from these results and the data from the animal and in vitro work to make them relevant.

Answer: *The major findings of this study are that early life stress mediated via NGF/TrkA signaling leads to the EC cell hyperplasia and increased serotonin production in the GI tract, resulting in IBS-like phenotypes including visceral hyperalgesia. Our results showing the significant positive correlation between serum NGF and serotonin in IBS-D patients and healthy controls can further strengthen our major conclusion and provides the physiological relevance with clinical implication for our study. Although we think this data does provide a solid support for our conclusion, we do not mind removing this data if the reviewer and the editor find it inappropriate for this manuscript.*

Due to the technical limitation, we cannot confirm our results obtained from NMS models in human in vivo. To enhance the human relevance of our project, we cultured human colonic organoids with or without recombinant human NGF. Consistent with the results obtained from mouse organoids, culture with rNGF significantly increased the size of human organoids (revised Supplementary Fig. 6a-b). Increased expression of LGR5 was detected in human organoids cultured with NGF (revised Supplementary Fig. 6c), suggesting that NGF enhances the self-renewal of human colonic stem cells in organoids. Furthermore, human organoids cultured with rNGF exhibit increased density of serotonin-producing EC cells (revised Supplementary Fig. 6d-e). Overall, we found that NGF promotes the self-renewal of stem cells and their differentiation into serotonin-producing EC cells in both mouse and human organoids.

7. Other points - are the doses of NGF used pharmacological or biological?

Answer: The dose of NGF used in the experiment is 1 μ g/kg. We examined the concentration of plasma NGF in mice at 3 hours after receiving NGF injection by ELISA. Injection of recombinant NGF resulted in a nearly 2.5-fold increase in plasma NGF (from 7.379pg/ml to 18.368pg/ml, pls. see below). Similarly, the plasma NGF concentration in mice was dramatically increased by more than 3 folds upon maternal deprivations (from 8.834pg/ml to 28.007pg/ml, pls. see below). Therefore, the dose of NGF used in the experiment is likely within the biological range.

Page 5, line 12 - this sentence is incomplete

Answer: We are sorry for the mistake and have revised the sentence in the manuscript. (pls. see P.5 lane 13-15)

Page 5, para 3 - what is the evidence that early-life stress "alters the specification of intestinal cell lineages"?

Answer: NMS leads to increased density of both EC and Paneth cells in the gut, but it does not alter the Goblet cell. These results suggest that it affects the specification of intestinal cell lineages.

Reviewer #3:

This manuscript describes an interesting and novel idea; that early life stress (modeled by maternal separation) can cause later symptoms similar to IBS by working through the NGF-trkA axis, and that this then trans activates the Wnt signaling cascade to modify stem cells. Unfortunately this work suffers from two main deficiencies: the first is that the experiments are described in insufficient detail to actually allow an understanding of what might actually be going on in the system, and second and more importantly, they use reagents of insufficient specificity to actually back up the claims.

Thank you so much for the positive comments and insightful criticisms for improving our manuscript.

To discuss the second point first, the paper relies heavily on the use of K252a as an inhibitor of trkA. K252a does inhibit trkA, but it is not specific, nor even very selective, inhibiting a wide variety of tyrosine kinases and even serine-threonine kinases. This has been known for quite some time (Ruegg and Burgess, Trends in Pharmaceutical Science, 10, 218-220, 1989; Mizuno, et al., Federation of European Biochemical Societies, 330, 114-116, 1993; Martin et al., Neuropharmacology 61, 148-155, 2011 as examples). Use of such a non-selective inhibitor, especially with no information given about the in vivo dose used, the dosing schedule or the timing of the experiments, much less the actual exposure in vivo with assays of drug levels, can tell almost nothing about the proposed involvement of the NGF-trkA system.

Answer: *Following the reviewers' suggestion, we repeated the experiments by using NMAC13, a well characterized anti-TrkA monoclonal antibody with remarkable neutralizing properties^{2,3}. Consistent with the results obtained from the experiments using K252a, we found that blocking TrkA with NMAC13 effectively suppresses the expansion of Lgr5⁺ colonic stem cells and reduces the number of serotonin-producing EC cell in NMS mice (revised Supplementary Fig 2a-b). Furthermore, inhibiting TrkA also reduced the colonic serotonin content in NMS mice to a level similar to that of the control mice (revised Supplementary Fig 2c). We provided the experimental procedures for the drug treatment in the Method and Material of the original manuscript (pls. see P.13 lane 14-22). However, as it may not be detailed enough, we now revised the protocol and added a diagram showing the timeframe of animal experiments in the Supplementary Fig. 12 (pls. also see below).*

Second, throughout the paper, there is insufficient methodological detail to understand what the authors are proposing. They show that neonatal separation can result in a variety of effects much later in life and then state that NGF administration can mimic these. But they never address how much NGF is administered, when it is administered,

how often or for how long it is administered or how long after administration the various assays take place. The same is true about the experiments with administration of purported NGF inhibitors, when are they administered, at what dose, how often, for how long and how soon before assays take place? Without this information, it is not possible to try to understand what might be going on in the animal.

Answer: *We are sorry for the insufficient information in the manuscript. We now revised the Method and Material by providing a more detailed protocol for the drug administration (Pls. see P.13 lane 14-22; Supplementary Fig. 12).*

Does neonatal separation induce an NGF and/or trkA upregulation that lasts for life and needs to be acutely and continuously inhibited for cessation of the effects? Or is inhibition some sort of breaking the cycle that leads to a long-term cessation?

Answer: *Due to the time restriction, we cannot examine the life-long impact of NMS on the expression of NGF and TrkA within the period of revision. In **Supplementary Fig. 1a-b**, we showed the upregulation of NGF and TrkA in the colon of NMS rats at 8 weeks upon maternal deprivation. In consistence with our study, another study also reported the upregulation of NGF and TrkA at 14 days and 12 weeks of age for NMS rat⁴. Therefore, NGF/TrkA likely maintains long-term alterations of intestinal integrity induced by NMS.*

*Our data showed that mice receiving the treatment of recombinant NGF during postnatal day 3-14 exhibit various IBS-like pathophysiological phenotypes that resemble to those observed in NMS-treated mice (**Fig. 1-2, supplementary Fig. 1-5**). Therefore, short-term exposure to stressful condition (or high level of NGF) during the early life is enough to increase the susceptibility of animals to GI functional diseases in the later life. Besides, we also observed that blocking NGF/TrkA by daily administration of inhibitors during NMS treatment effectively ameliorates the NMS-induced intestinal changes, such as expansion of stem cells/Paneth cells/EC cells mice (**Fig. 1-2, Supplementary Fig. 1-5**). More importantly, inhibiting TrkA also suppresses the upregulation of NGF induced by NMS in murine colons (**revised Supplementary Fig. 2d**). These findings collectively suggest that NGF inhibition during NMS treatment can lead to a long-term cessation of the NMS effects.*

And even more basically, what is the effective dose of the non-selective inhibitor K252a?

Answer: *We tested the effective dose of K252a in our previous preliminary studies. The effective dose of K252a was determined by assessing the dose of K252a for inhibition of NGF-induced proliferation of CaCo2 cells. In these experiments, NGF was used at the dose of 100ng/ml. IC₅₀ for K252a was determined at 27.4ng/ml. We*

used 100-fold higher doses for in vivo experiments. K252a was used at the dose of 25, 50, and 100µg/kg in mice to determine its tolerability. The doses of 25 and 50 µg/kg were well tolerated without side effects. We therefore chose 50 µg/kg for further experiments.

This system is of obvious interest but there are other issues with the work in this paper (the in vitro dose of NGF used is exceptionally high and calls into question the relevance of any effects seen).

Answer: *According to the suggestion by protein manufacturer ThermoFisher, the biological effects of NGF 2.5S in vitro are usually observed at 1 to 100 ng/ml (0.2 to 4nM). The recommended concentration to be used in vitro for maintenance of sympathetic and sensory nerve cultures is 50 ng/mL of medium. Based on this suggestion, we have tested the effect of NGF with different doses (1, 10, 50, 100 and 500 ng/ml) in organoid culture. We found that the growth-promoting effects of NGF on organoid culture could be observed at 10ng/ml and achieved maximum at 100ng/ml. Therefore, we chose 100ng/ml for further experiments.*

Following the reviewer's advices, we cultured human colonic organoids with recombinant NGF at the dose of 10ng/ml. NGF treatment did not only promote the growth of human colonic organoids, but also resulted in the enrichment of serotonin-producing EC cells in the organoids (revised Supplementary Fig. 6).

The standard culture medium for intestinal organoid formation usually consists of a cocktail of growth factors including recombinant murine EGF (50ng/ml), recombinant human R-spondin 1 (500ng/ml) and recombinant murine Noggin (50ng/ml)⁵. The concentration of NGF used in our experiment is indeed similar to that of other standard factors. The reason for high concentration of growth factors to be used for maintenance of organoid culture may be that 3-D organoids are embedded in Matrigel which significantly reduces the amount of growth factors that can reach the organoid.

More importantly, the results from mouse organoids are largely consistent with those obtained from the animal model treated with NGF/NMS, revealing the physiological relevance of our organoid-based studies.

Overall, the fact that the interpretations in this work depend so heavily on K252a and that the experiments are insufficiently described makes it impossible to recommend this paper for publication.

References

1 Cremon, C. et al. Intestinal serotonin release, sensory neuron activation, and

- abdominal pain in irritable bowel syndrome. Am J Gastroenterol* **106**, 1290-1298, doi:10.1038/ajg.2011.86 (2011).
- 2 *Cattaneo, A. et al. Functional blockade of tyrosine kinase A in the rat basal forebrain by a novel antagonistic anti-receptor monoclonal antibody. J Neurosci* **19**, 9687-9697 (1999).
- 3 *Ugolini, G., Marinelli, S., Covaceuszach, S., Cattaneo, A. & Pavone, F. The function neutralizing anti-TrkA antibody MNAC13 reduces inflammatory and neuropathic pain. Proc Natl Acad Sci U S A* **104**, 2985-2990, doi:10.1073/pnas.0611253104 (2007).
- 4 *Barreau, F., Cartier, C., Ferrier, L., Fioramonti, J. & Bueno, L. Nerve growth factor mediates alterations of colonic sensitivity and mucosal barrier induced by neonatal stress in rats. Gastroenterology* **127**, 524-534 (2004).
- 5 *Sato, T. et al. Single Lgr5 stem cells build crypt-villus structures in vitro without a mesenchymal niche. Nature* **459**, 262-265, doi:10.1038/nature07935 (2009).

Reviewers' comments:

Reviewer #1 (Remarks to the Author):

Authors lately addressed my concerns.

Reviewer #2 (Remarks to the Author):

I am satisfied with the authors' responses to my comments.

Reviewer #4 (Remarks to the Author):

Reviewer #3 is quite correct in their concern about the authors' reliance on K252a as the Trk inhibitor. Many used this compound in the early days of Trk signalling research, until it was supplanted by much more selective compounds. Cephalon, which characterized K252a, subsequently generated Lestaurtinib (CEP 701), which was more selective, and particularly no longer had the JNK inhibitory activity that K252a has. The third generation of Trk inhibitors include GW441756 and LOXO-101, even more potent and selective. To rely on K252a for almost all experiments (in 4 of the main figures), especially without biochemical or genetic validation that activated TrkA is a target, puts the entire study in doubt. Required are experiments demonstrating that K252a and at least one other more modern inhibitor are working on target and are reasonably selective, and genetic studies that show that TrkA knockdown or knockout has a similar phenotype to K252a.

To address the reviewer's concerns, the authors added in two suppl figures (2 and 9) a second Trk inhibitor, a neutralizing TrkA antibody NMAC13 that more or less phenocopies some of the results using K252a. However, as described below, this does not satisfy me (see below).

Here are the ongoing issues, which should have been addressed in the first round to answer Reviewer #3.

1. No where in the paper do the authors show that K252a or NMAC13 are working on target. This is easy to do, as there are excellent phospho-Trk antibodies that work both in Western blots and for IHC/IF. Just treat with the inhibitor, lyse the cells, organoids or tissues, and probe in western blots with anti-P-TrkA and reprobe with anti-TrkA.

2. A second control that should have been performed is to show that K252a and NMAC13 inhibit the phenotypic changes induced by NGF.

3. There is one total TrkA protein blot in the study (Supp Fig. 1B), and I have my doubts about it. TrkA is typically a fuzzy wide band due to extensive glycosylation of the extracellular domain, not a sharp band as shown. The actin loading control is overloaded so the blot cannot be quantified.

4. NMAC13 is not well-characterized, as it has not been shown to be specific for inhibiting TrkA and not TrkB and C. In the two papers that used this antibody to probe TrkA function, neither examined whether it inhibited TrkA phosphorylation in primary cells.

5. Genetic confirmation is really needed showing that TrkA activity is responsible for expanding the intestinal compartment. There are several ways to do this. Trk floxed mice are available, and an intestinal-specific Cre-ERT2 would have readily confirmed the many interesting findings. Even better to use are the TrkA (F592A) knockin mice of David Ginty (Chen et al Neuron 2005). These mice, which Ginty provides to those who ask, encode TrkA that is susceptible to inhibition by 1NMPP1. Administration to cells, organoids or mice very specifically will inhibit TrkA.

Editorial Note: Further comments were obtained from reviewer #4 as part of discussions between the editorial team, the authors, and reviewer #4 regarding the revision of this manuscript. These comments are pasted below.

Reviewer #4 (Further remarks to the Author):

The major issue with this report is the reliance on K252a as the main loss-of-function reagent to assess the role of the NGF-Trk signalling axis. K252a is a non-selective Trk inhibitor that has many targets besides TrkA and has been supplanted by much better inhibitors. For journals such as this, the main reagent should have been NMAC13, with additional and crucial validation of its effects that was not done in the few papers that have been published using this theoretically much more selective Trk inhibitor. Instead, the authors have used it in a few supplementary figures as the confirming loss-of-function reagent. My recommendation is that the authors perform additional experiments using NMAC13 as the lead NGF-TrkA signalling inhibitor, adding validation in the suppl. figures that the reagent is working on-target (inhibiting p-TrkA by IHC or western blot) and that it inhibits NGF-induced biological responses, importantly both in their in vitro and in vivo systems.

Again, I would have readily accepted the authors' findings if they had used MNAC13 as their primary reagent, substantiating that it inhibited p-TrkA in vitro and in vivo in the intestinal cells and not other Trks, and that it inhibited the phenotypic changes induced by NGF. K252, however, is the lead compound, and its broad inhibitory actions on many kinases including JNK make it very problematic, even for proof-of-principle studies. It just cannot be trusted now as the primary Trk inhibitory reagent for a journal such as this one, especially when better and more selective inhibitors and genetic techniques are available. This is why I suggested the pharmacogenetic approach of the Ginty lab mice that a number in the field have used to substantiate their Trk inhibitor results. I realize, however, that doing the "right" confirming experiments with genetic models (one of NGF, Trk knockouts, Ginty lab TrkA knocking mice or even TrkA RNAi), would take a year or more to ship, re-derive and do the experiments on these mice. I will instead accept more NMAC13 experiments, presented in the main figures instead of the suppl data, and with experiments showing that the reagent works on-target in the authors' biological systems. The K252a experiments could be moved to suppl data.

Let's deal with the contention that NMAC13 is "well-characterized". The function-blocking TrkA antibody generated by the Cattaneo lab has been used to assess NGF-TrkA function in just a few papers. In Cattaneo et al J. Neurosci 1999, the antibody blocked NGF binding to TrkA, and bound TrkA much more efficiently than TrkB. It was not tested against TrkC. In IHC, it gave different staining than anti-TrkC. It was assessed biologically in two systems but not biochemically. Ugolini et al PNAS 2007 did assess whether it would block the tyrosine phosphorylation of TrkA in 3T3 cells exogenously expressing it. 3T3 cells expressing TrkB and TrkC were not assessed, nor was this attempted in primary cells. Pasavento et al Neuron 2000 used the very specific TrkA-IgG fusion that binds NGF in addition to the anti-TrkA to validate their results. Again, biochemistry was not attempted. I agree that the antibody has been well-characterized for TrkA binding and function-blocking, but not biochemically. However, as for all reagents such as this, the confidence of the field to use them (in addition to the three labs in Italy) and for myself regarding the findings of this paper are based upon whether in the authors' particular system the reagent can be shown to be very selective at inhibiting TrkA. The antibody could bind similar epitopes on other proteins, or differential glycosylation (which makes TrkA a broader band on western blots of neurons) could block epitopes as has been shown with several antibodies to the Trk extracellular domain. The usual experiments are to test the antibodies biochemically on 3T3 cells expressing TrkA,

TrkB or TrkC, show biochemically that it inhibits TrkA phosphorylation (using western blots with anti-phospho-Trk) and downstream signalling and biological effects in primary neurons expressing TrkA, and show minimal immunostaining using knockout models. Best is that phosphoproteomics be performed +/- NMAC13 and +/-NGF treatment to determine if there are off-target effects. There has been a paucity of signalling data validating this antibody, which likely explains what it has not been extensively used. Therefore, it is now up to the authors to perform these experiments (not phosphoproteomics) so that the field can trust the results.

Minor point. I can accept that the sharp band is TrkA in intestinal cells, if the band in question is the same when reprobed with anti-phospho-TrkA. Not the usual band, as TrkA is heavily glycosylated giving a characteristic broad 140kD band and an underglycosylated 85-90kD band in many cell types. The Santa Cruz catalogue blot is not from a reviewed paper and is from overexpressing cells. Reviewers of neurotrophin papers do question when we see sharp TrkA bands, but this appearance may be tissue-specific, and also result from the use of gradient minigel formats that can sharpen the Trk bands.

Reply to reviewers' comments

The major issue with this report is the reliance on K252a as the main loss-of-function reagent to assess the role of the NGF-Trk signalling axis. K252a is a non-selective Trk inhibitor that has many targets besides TrkA and has been supplanted by much better inhibitors. For journals such as this, the main reagent should have been MNAC13, with additional and crucial validation of its effects that was not done in the few papers that have been published using this theoretically much more selective Trk inhibitor. Instead, the authors have used it in a few supplementary figures as the confirming loss-of-function reagent. My recommendation is that the authors perform additional experiments using MNAC13 as the lead NGF-TrkA signalling inhibitor, adding validation in the suppl. figures that the reagent is working on-target (inhibiting p-TrkA by IHC or western blot) and that it inhibits NGF-induced biological responses, importantly both in their in vitro and in vivo systems. Again, I would have readily accepted the authors' findings if they had used MNAC13 as their primary reagent, substantiating that it inhibited p-TrkA in vitro and in vivo in the intestinal cells and not other Trks, and that it inhibited the phenotypic changes induced by NGF.

*Answer: We really appreciate the constructive comments and insightful criticisms from the reviewer. Following the suggestions of the reviewer, we have performed additional experiments using MNAC13 as a major inhibitor of NGF/TrkA signaling. Most of the data involving K252a in both main and supplementary figures has been replaced with the new data using MNAC13. In the new data set, we showed that either NGF treatment or NMS leads to the increased phosphorylation of TrkA in the proximal colon (**revised figure 1b**), the EC cell hyperplasia along with increased colonic serotonin production (**revised figure 1a-d**;) as well as the expansion in stem cell and Paneth cell compartment in mice (**revised figure 2b-e**; **supplementary figure 3 & 5**). These phenotypic changes induced by NGF or NMS were completely blocked by inhibiting TrkA with MNAC13. Consistent with in vivo data, we also found that blockade of TrkA by MNAC13 effectively inhibited the phosphorylation of TrkA induced by recombinant NGF and the NGF-promoted growth in intestinal organoids (**revised supplementary figure 6d-f**). These new results are largely consistent with the previous data using K252a.*

K252, however, is the lead compound, and its broad inhibitory actions on many kinases including JNK make it very problematic, even for proof-of-principle studies. It just cannot be trusted now as the primary Trk inhibitory reagent for a journal such as this

one, especially when better and more selective inhibitors and genetic techniques are available. This is why I suggested the pharmacogenetic approach of the Ginty lab mice that a number in the field have used to substantiate their Trk inhibitor results. I realize, however, that doing the “right” confirming experiments with genetic models (one of NGF, Trk knockouts, Ginty lab TrkA knocking mice or even TrkA RNAi), would take a year or more to ship, re-derive and do the experiments on these mice. I will instead accept more NMAC13 experiments, presented in the main figures instead of the suppl data, and with experiments showing that the reagent works on-target in the authors’ biological systems. The K252a experiments could be moved to suppl data.

Answer: Thank you very much for the reviewer’s understanding on our difficulties in performing experiments using the new animal model. As we agree with the reviewer that K252a is not a specific inhibitor of TrkA, we therefore replaced all experiments using K252a with a new set of data set using MNAC13 as the major inhibitor for TrkA.

Let’s deal with the contention that NMAC13 is “well-characterized”. The function-blocking TrkA antibody generated by the Cattaneo lab has been used to assess NGF-TrkA function in just a few papers. In Cattaneo et al J. Neurosci 1999, the antibody blocked NGF binding to TrkA, and bound TrkA much more efficiently than TrkB. It was not tested against TrkC. In IHC, it gave different staining than anti-TrkC. It was assessed biologically in two systems but not biochemically. Ugolini et al PNAS 2007 did assess whether it would block the tyrosine phosphorylation of TrkA in 3T3 cells exogenously expressing it. 3T3 cells expressing TrkB and TrkC were not assessed, nor was this attempted in primary cells. Pasavento et al Neuron 2000 used the very specific TrkA-IgG fusion that binds NGF in addition to the anti-TrkA to validate their results. Again, biochemistry was not attempted. I agree that the antibody has been well-characterized for TrkA binding and function-blocking, but not biochemically. However, as for all reagents such as this, the confidence of the field to use them (in addition to the three labs in Italy) and for myself regarding the findings of this paper are based upon whether in the authors’ particular system the reagent can be shown to be very selective at inhibiting TrkA. The antibody could bind similar epitopes on other proteins, or differential glycosylation (which makes TrkA a broader band on western blots of neurons) could block epitopes as has been shown with several antibodies to the Trk extracellular domain. The usual experiments are to test the antibodies biochemically on 3T3 cells expressing TrkA, TrkB or TrkC, show biochemically that it inhibits TrkA phosphorylation (using western blots with anti-phospho-Trk) and downstream signalling and biological effects in primary neurons expressing TrkA, and show minimal immunostaining using knockout models. Best is that phosphoproteomics be performed +/- NMAC13

and +/-NGF treatment to determine if there are off-target effects. There has been a paucity of signalling data validating this antibody, which likely explains what it has not been extensively used. Therefore, it is now up to the authors to perform these experiments (not phosphoproteomics) so that the field can trust the results.

Answer: Thank you for the reviewer's suggestions. We obtained 3T3 cells expressing TrkA/B/C from our collaborator, Prof. Frank M. Longo, and performed the suggested experiments. MNAC13 effectively inhibited the NGF-induced phosphorylation of TrkA and Akt in 3T3 cells expressing TrkA. However, it did not suppress the phosphorylation of TrkB/C induced by BDNF and NT-3 in 3T3 cells expressing TrkB/C (pls. see the figure below).

To further validate the function of MNAC13 in primary cells within the scope of this study, we examined the effect of MNAC13 in intestinal organoid culture. The blockade of TrkA by MNAC13 did not only inhibit the phosphorylation of TrkA induced by NGF, but also suppressed the NGF-promoted growth in intestinal organoids (revised supplementary figure 6d-f). In addition, we failed to detect TrkB and TrkC in the organoids by Western blotting, which is in consistency with previous studies showing that TrkB and TrkC are just minimally expressed in non-neuronal tissues, such as normal colons (Kim et al., 2017; Otani et al., 2017). Our new data obtained from both in vitro and in vivo experiments along with the previous publications describing MNAC13 have already yielded sufficiently convincing and physiologically-relevant results showing that MNAC13 is a specific inhibitor of TrkA at least in our system.

Minor point. I can accept that the sharp band is TrkA in intestinal cells, if the band in question is the same when reprobred with anti-phospho-TrkA. Not the usual band, as TrkA is heavily glycosylated giving a characteristic broad 140kD band and an underglycosylated 85-90kD band in many cell types. The Santa Cruz catalogue blot is not from a reviewed paper and is from overexpressing cells. Reviewers of neurotrophin papers do question when we see sharp TrkA bands, but this appearance may be tissue-specific, and also result from the use of gradient minigel formats that can sharpen the Trk bands.

Answer: We examined the total and phosphorylated forms of TrkA in both colon tissues and intestinal organoids and consistently detected a sharp band of TrkA at about 140kDa, which is consistent with the results of other previous studies using the same antibody detecting TrkA [Examples:(Culmsee et al., 2002; Lambiase et al., 2005; Li et al., 2009; Perrone et al., 2005; Zhang and Chen, 2007)]. Similarly, our collaborator, Prof. Longo, also detected a sharp band of Trk-A in 3T3 cells expressing TrkA [Figure below adopted from (Yang et al., 2016)]

Moreover, a previous study reporting the expression of TrkA in human small intestines consistently showed a single shape band of TrkA [Figure below adopted from (di Mola et al., 2000)].

With the support of data from various published studies, we believe that the variation in the appearance of TrkA band is likely resulted from the difference in antibodies used in the experiment and the type of tissues examined.

Reviewer #4 (Remarks to the Author):

Reviewer #3 is quite correct in their concern about the authors' reliance on K252a as the Trk inhibitor. Many used this compound in the early days of Trk signalling research, until it was supplanted by much more selective compounds. Cephalon, which characterized K252a, subsequently generated Lestaurtinib (CEP 701), which was more selective, and particularly no longer had the JNK inhibitory activity that K252a has. The third generation of Trk inhibitors include GW441756 and LOXO-101, even more potent and selective. To rely on K252a for almost all experiments (in 4 of the main figures), especially without biochemical or genetic validation that activated TrkA is a target, puts the entire study in doubt. Required are experiments demonstrating that K252a and at least one other more modern inhibitor are working on target and are reasonably selective, and genetic studies that show that TrkA knockdown or knockout has a similar phenotype to K252a.

To address the reviewer's concerns, the authors added in two suppl figures (2 and 9) a second Trk inhibitor, a neutralizing TrkA antibody MNAC13 that more or less phenocopies some of the results using K252a. However, as described below, this does not satisfy me (see below).

Here are the ongoing issues, which should have been addressed in the first round to answer Reviewer #3.

Answer: Thank you so much for the detailed review and insightful comments from the reviewer. In general, we agree with the comments given by the reviewer. We performed additional experiments to validate the specificity of MNAC13. In addition, we replaced the data involving K252a with a new set of data using MNAC13 as a primary inhibitor of TrkA. For details, pls see our replies below.

1. No where in the paper do the authors show that K252a or MNAC13 are working on target. This is easy to do, as there are excellent phospho-Trk antibodies that work both in Western blots and for IHC/IF. Just treat with the inhibitor, lyse the cells, organoids or tissues, and probe in western blots with anti-P-TrkA and reprobe with anti-TrkA.

*Answer: We thank the reviewer for raising this important issue. To address this issue, we examined the phosphorylation of TrkA in NGF-stimulated organoids treated with/without TrkA inhibitors and in colonic tissues derived from NMS/NGF mice treated with/without TrkA inhibitors by western blotting. We found that the phosphorylation of TrkA increased in colonic tissues derived from NMS/NGF treated mice, which was completely inhibited by the treatment of MNAC13 (**revised Figure 1b**). In addition, blocking TrkA with MNAC13 also inhibited the*

phosphorylation of TrkA induced by NGF in the intestinal organoids (revised supplementary Figure 6d-f).

2. A second control that should have been performed is to show that K252a and NMAC13 inhibit the phenotypic changes induced by NGF.

Answer: In the revised manuscript we showed that inhibiting of TrkA by the treatment MNAC13 effectively suppressed the phenotypic changes induced by NGF including EC cell hyperplasia along with increased serotonin production (revised Figure 1), the expansion of colonic and intestinal stem cell compartments (revised Figure 2 & Supplementary Figure 5), the elevated density of Paneth cells (revised Supplementary Figure 3) and the increased phosphorylation of TrkA and β -catenin in the colonic tissues (revised Figure 1 and Figure 4).

3. There is one total TrkA protein blot in the study (Supp Fig. 1B), and I have my doubts about it. TrkA is typically a fuzzy wide band due to extensive glycosylation of the extracellular domain, not a sharp band as shown. The actin loading control is overloaded so the blot cannot be quantified.

Answer: For the actin loading control in Supplementary Fig. 1B, we have provided a less exposed film and the expression of TrkA was quantified again. We consistently showed that the expression of TrkA was significantly upregulated in the colonic tissues of rat challenged by NMS. For the appearance of bands, please see our reply to the comment above.

4. NMAC13 is not well-characterized, as it has not been shown to be specific for inhibiting TrkA and not TrkB and C. In the two papers that used this antibody to probe TrkA function, neither examined whether it inhibited TrkA phosphorylation in primary cells.

Answer: Please see our replies to the comments above.

5. Genetic confirmation is really needed showing that TrkA activity is responsible for expanding the intestinal compartment. There are several ways to do this. Trk floxed mice are available, and an intestinal-specific Cre-ERT2 would have readily confirmed the many interesting findings. Even better to use are the TrkA (F592A) knockin mice of David Ginty (Chen et al Neuron 2005). These mice, which Ginty provides to those who ask, encode TrkA that is susceptible to inhibition by 1NMPP1. Administration to cells, organoids or mice very specifically will inhibit TrkA.

Answer: We have now performed additional experiments to validate the specificity of MNAC13. In addition, we repeated most of the previous in vitro and in vivo experiments with MNAC13 as the major inhibitor of TrkA and consistently showed that blocking TrkA with MNAC13 effectively suppressed the phenotypic changes induced by NMS/NGF treatment. We hope that our experiments using inhibitors promise clinical relevance and potential as much as the genetic model does.

References

- Culmsee, C., N. Gerling, M. Lehmann, M. Nikolova-Karakashian, J.H. Prehn, M.P. Mattson, and J. Kriegstein. 2002. Nerve growth factor survival signaling in cultured hippocampal neurons is mediated through TrkA and requires the common neurotrophin receptor P75. *Neuroscience*. 115:1089-1108.
- di Mola, F.F., H. Friess, Z.W. Zhu, A. Koliopoulos, T. Bley, P. Di Sebastiano, P. Innocenti, A. Zimmermann, and M.W. Buchler. 2000. Nerve growth factor and Trk high affinity receptor (TrkA) gene expression in inflammatory bowel disease. *Gut*. 46:670-679.
- Kim, M.S., K.W. Suh, S. Hong, and W. Jin. 2017. TrkC promotes colorectal cancer growth and metastasis. *Oncotarget*. 8:41319-41333.
- Lambiase, A., D. Merlo, C. Mollinari, P. Bonini, A.M. Rinaldi, D.A. M, A. Micera, M. Coassin, P. Rama, S. Bonini, and E. Garaci. 2005. Molecular basis for keratoconus: lack of TrkA expression and its transcriptional repression by Sp3. *Proc Natl Acad Sci U S A*. 102:16795-16800.
- Li, H., C. Costantini, H. Scrabble, R. Weindruch, and L. Puglielli. 2009. Egr-1 and Hipk2 are required for the TrkA to p75(NTR) switch that occurs downstream of IGF1-R. *Neurobiol Aging*. 30:2010-2020.
- Otani, K., M. Okada, and H. Yamawaki. 2017. Diverse distribution of tyrosine receptor kinase B isoforms in rat multiple tissues. *J Vet Med Sci*. 79:1516-1523.
- Perrone, L., S. Paladino, M. Mazzone, L. Nitsch, M. Gulisano, and C. Zurzolo. 2005. Functional interaction between p75NTR and TrkA: the endocytic trafficking of p75NTR is driven by TrkA and regulates TrkA-mediated signalling. *Biochem J*. 385:233-241.
- Yang, T., S.M. Massa, K.C. Tran, D.A. Simmons, J. Rajadas, A.Y. Zeng, T. Jang, S. Carsanaro, and F.M. Longo. 2016. A small molecule TrkB/TrkC neurotrophin receptor co-activator with distinctive effects on neuronal survival and process outgrowth. *Neuropharmacology*. 110:343-361.

Zhang, J., and X. Chen. 2007. *DeltaNp73 modulates nerve growth factor-mediated neuronal differentiation through repression of TrkA*. *Mol Cell Biol.* 27:3868-3880.

Reviewers' comments:

Reviewer #4 (Remarks to the Author):

The authors have now satisfactorily responded to my comments. They have redid key experiments using a much more selective TrkA inhibitor, and the results are similar to those using K252a. I do ask that the authors include the validation experiments showing that MNAC13 suppresses TrkA and not TrkB or TrkC activity in 3T3 cells, shown on page 4 of the rebuttal. This data gives the field confidence that this rarely-used reagent is specifically targeting TrkA. It can be inserted into Supp. Fig. 3, as panel d.

Reviewer #4 (Remarks to the Author):

The authors have now satisfactorily responded to my comments. They have redid key experiments using a much more selective TrkA inhibitor, and the results are similar to those using K252a. I do ask that the authors include the validation experiments showing that MNAC13 suppresses TrkA and not TrkB or TrkC activity in 3T3 cells, shown on page 4 of the rebuttal. This data gives the field confidence that this rarely-used reagent is specifically targeting TrkA. It can be inserted into Supp. Fig. 3, as panel d.

*Answer: Thank you for the suggestion. We have put the results validating the specificity of MNAC13 in **Supplementary Fig. 3**.*